# SCALING LAWS FOR NEURAL MACHINE TRANSLATION

**Behrooz Ghorbani,**[*] **Orhan Firat, Markus Freitag, Ankur Bapna, Maxim Krikun,**
**Xavier Garcia, Ciprian Chelba & Colin Cherry**
Google AI

## ABSTRACT

We present an empirical study of the scaling properties of encoder-decoder Transformer models used in neural machine translation (NMT). We show that cross-entropy loss as a function of model size follows a certain scaling law. Specifically (i) We propose a formula which describes the scaling behavior of cross-entropy loss as a bivariate function of encoder and decoder size, and show that it gives accurate predictions under a variety of scaling approaches and languages; we show that the total number of parameters alone is not sufficient for such purposes. (ii) We observe different power law exponents when scaling the decoder vs the encoder, and provide recommendations for optimal allocation of encoder/decoder capacity based on this observation. (iii) We also report that the scaling behavior of the model is acutely influenced by *composition bias* of the train/test sets, which we define as any deviation from naturally generated text (either via machine generated or human translated text). We observe that natural text on the target side enjoys scaling, which manifests as successful reduction of the cross-entropy loss. (iv) Finally, we investigate the relationship between the model size, cross-entropy loss and the quality of the generated translations. We find two different behaviors, depending on the nature of the test data. For test sets which were originally translated from target language to source language, both loss and BLEU score improve as model size increases. In contrast, for test sets originally translated from source language to target language, the loss improves, but the BLEU score stops improving after a certain threshold. We release generated text from all models used in this study.

## 1 INTRODUCTION

Scaling properties of neural networks have long been an intriguing topic of study (Ahmad & Tesauro, 1988; Amari et al., 1992). Along with the practical success of modern neural networks at scale, theoretical understanding of the factors governing the quality and training dynamics of large neural networks has also being developing (Advani et al., 2020; Rosenfeld et al., 2019; Geiger et al., 2020; Ghorbani et al., 2021; Cho, 2020; Hutter, 2021; Bahri et al., 2021; Loureiro et al., 2021). In particular, scaling model sizes, datasets and the total computation budget has been identified as a reliable approach to improve generalization performance on several machine learning tasks. For many of these tasks the scaling behavior of neural networks is highly predictable; model fit or test loss can be described precisely as a function of its number of parameters (Hestness et al., 2017; Kaplan et al., 2020; Henighan et al., 2020; Hernandez et al., 2021; Rosenfeld et al., 2019). Neural machine translation (NMT) has long enjoyed the benefits of scaling (Huang et al., 2019; Arivazhagan et al., 2019; Lepikhin et al., 2021), but studies investigating the scaling behavior of NMT models are missing. We present a large-scale systematic study of scaling laws for encoder-decoder Transformer models applied to NMT (Vaswani et al., 2017).

We start with highlighting the major differences between decoder-only language models, where the majority of the previous work has focused, and encoder-decoder (conditional) language models applied to NMT. The two differ along a few crucial dimensions. The first difference results from the very nature of the separate architectures being used, i.e. decoder-only vs encoder-decoder. The presence of separate architectural components complicates the study of scaling properties due to the

---

[*]Correspondence to `ghorbani@google.com`.

increased degree of freedom. Second, contrary to language modeling, the task of machine translation is conditional: the task is predictive rather than fully generative. Furthermore, this prediction task is ambiguous: there is no one right answer for a given source, and translations can vary substantially depending on the translator's incentives. This manifests itself as different scaling benefits for different test sets. To take an extreme example, a test set translated by someone who writes nearly word-for-word translations may benefit less from model scaling than one translated by someone who considers each translation a work of art. In this work, these differences in difficulty coincide with the translation direction of the test set; that is, whether the source was translated into the target (source-original) or vice versa (target-original). Source-original data has translated text on the target side, which contains several artifacts of "translationese" that distinguish it from text originally written in that language, often lacking the diversity and complexity of "natural" text (Koppel & Ordan, 2011), while target-original data requires the prediction of more complex natural text on the target side. Finally, unlike language models, NMT is evaluated on metrics that quantify generation quality against reference translations (for eg. BLEU) (Papineni et al., 2002) instead of evaluating model fit (perplexity) on an evaluation set.

In this paper, we aim to provide empirical answers to the following research questions:

1. **Does the encoder-decoder architecture for NMT share the same scaling law function as language models?** Contrary to previous work on LM, we show that a univariate law depending on the total number of parameters in the network does not adequately describe the scaling behavior of NMT models. Our scaling laws parameterize the cross entropy loss as a bivariate function of the number of encoder parameters and the number of decoder parameters as separate variables. Our results indicate that the scaling behavior is largely determined by the total capacity of the model, and the capacity allocation between the encoder and the decoder.

2. **How does the naturalness of source/target side data affect scaling behavior?** We study the effect of naturalness of the source and the target text, both for training and evaluation. When evaluating with target side natural text, scaling the model capacity continues improving model quality throughout our range of measurements. On the other hand, improvements on cross-entropy saturate (or reaches the irreducible error region) on source side natural evaluation sets even for moderately-sized models.

3. **Do scaling improvements in cross-entropy translate into corresponding improvements in generation quality?** Finally we study the relationship between generation quality and cross-entropy and how their correlation changes as we: (i) Scale different components of the model (encoder vs decoder) and (ii) Evaluate on source-natural or target-natural test sets.

Our results on multiple language pairs and training/test data compositions validate that **model scaling predictably improves the cross-entropy on validation data**. However, our findings also raise several questions regarding the effect of naturalness of training and evaluation text and how cross-entropy eventually relates with generation quality for auto-regressive generative models. [1]

## 2 EFFECT OF SCALING ON CROSS-ENTROPY

### 2.1 EXPERIMENTAL SETTING

**Model Architectures and Training** We train a series of pre-layer norm Transformer networks with varying sizes (Xiong et al., 2020). Models are trained with per-token cross-entropy loss and Adafactor optimizer (Shazeer & Stern, 2018). All models are trained with a fixed batch-size of 500k tokens and dropout rate of 0.1 for residuals, feed-forward activations and attention. All models are trained to near convergence for 500k training steps. Details of the model hyper-parameters are described in Appendix C.

**Model Scaling** Transformer architecture consists of Transformer Blocks: a cascade of self-attention, cross-attention and feed-forward layers, each having multiple adjustable hyper-parameters (e.g.

---

[1] A few weeks before the publication of this manuscript on Arxiv, Gordon et al. (2021) appeared on Open-Review. While both papers study scaling laws for NMT, our studies focus on different parameter regimes (393K-56M vs 100M-4B). See Appendix A for a comparison of the results.

model-dimension, number of attention heads, attention projection dimension etc.). Considering the combinatorial expansion of the search space for scaling each one, in this study we choose to vary only the total number of Transformer Blocks, while keeping the internal hyper-parameters intact across different scales. In other words, we scale the depth of the Transformers while keeping width and other variables fixed. [2] We use GPipe pipeline parallelism for our experiments (Huang et al., 2019).

In an encoder-decoder Transformer architecture for NMT, depth scaling can naturally be implemented by varying encoder-decoder blocks independently or symmetrically. Hence, we examine the change in the cross-entropy loss as the number of parameters increase with three depth scaling approaches:

*Encoder Scaling*: vary encoder depth (2 to 64) while decoder depth is fixed (6 layers).

*Decoder Scaling*: vary decoder depth (2 to 64) while encoder depth is fixed (6 layers).

*Symmetric Scaling*: increasing decoder and encoder layers together (from 2 to 64), i.e. the number of Transformer Blocks in the encoder and decoder being equal.

For all experiments, configuration of the individual layers is unchanged: the model dimension, width of the feed-forward layer, and number of attention heads are fixed respectively at $1024$, $8192$, and $16$. [3] Each encoder layer adds approximately 20M parameters to the model while each decoder layer adds around 25M parameters. We train 200 such models which scale the model size by approximately a factor of 50 (from 80M parameters to 4B parameters). Following the convention, we do not count the parameters in the embedding and softmax layers towards the model size. We focus our study on large-scale models: our smallest models require 200 TPUv3 days to train to convergence while our largest models require 2700 TPUv3 days of training.

**Language Pairs**   We report results on four language pairs: English→German, German→English, English→Chinese, and Chinese→English. We use in-house web-crawled training datasets with around 2.2 billion sentence pairs (approximately 55 billion tokens) for English↔German and 781 million sentence pairs for English↔Chinese.

**Evaluation Sets**   We use a variety of test sets for evaluation covering different domains: (i) Web-Domain (ii) News-Domain (iii) Wikipedia (iv) Patents. The news-domain test sets come from the WMT2019 (Barrault et al., 2019) evaluation campaign (newstest2019) for all language pairs. The other test sets are internal test sets representing the different domains, ranging from 500 to 5000 sentence pairs. For each domain, we randomly sample sentences in the source language and use professional translators to generate a reference translation in the target language. Throughout the paper, we will refer this type of test sets as *source-original* as the source sentences have been crawled from the web while the reference translations are added later. For most of the domains, we also have a *target-original* counterpart which is generated in the opposite direction: Sentences are crawled in the target language and human translated into the source language. Earlier work (Freitag et al., 2019; 2020; Graham et al., 2020) showed that it is important to differentiate between the two different kinds of test sets as the style of natural sentences and human (or machine) translations (translationese) is quite different. Cross-entropy loss is evaluated on the different test sets during training. To reduce the variation caused by the parameter fluctuations at the end of the training, we present the median loss over the last 50k steps of the training as the final loss.

## 2.2 RESULTS

Figure 1 shows the empirical evolution of the test loss on the Web-Domain test sets for encoder and decoder scaling for English→German. To compare the empirical results with the scaling laws present in the literature for decoder only models (Kaplan et al., 2020; Henighan et al., 2020), we have fitted a power law of the form $\hat{L}(N) = \alpha N^{-p} + L_\infty$ to the data. [4] Here, $N$ is the total number of parameters outside of embedding / softmax layers and $\{\alpha, p, L_\infty\}$ are fitted parameters. As Figure 1 suggests, scaling the encoder has different effects on the test loss compared to scaling the decoder. As such, simple power-laws that only consider the total parameter count, fail to capture the correct scaling behavior of the model.

---

[2]In Appendix H, we examine the role of scaling approach in our results. See Section 2 for more details.

[3]A complete description of the model architecture is provided in Appendix C

[4]Details of the curve fitting procedure are presented in Appendix I.

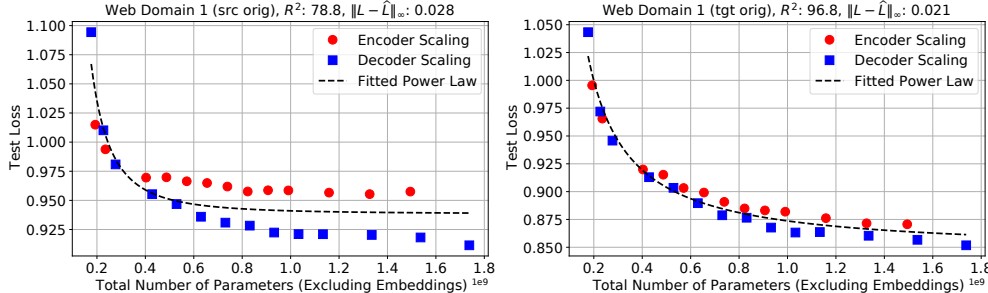

Figure 1: Evolution of the test loss as a function of the total model parameters for English→German. Scaling the encoder has different effects compared to scaling the decoder. As such, traditional scaling laws are unable to capture the correct scaling behavior. R-squared ($100 \times \frac{\text{explained variance}}{\text{total variance}}$) and maximum absolute deviation ($\| \cdot \|_\infty$) are reported for each fit.

**Proposed Scaling Law**    To tackle this issue, we present a new scaling law that reflects the encoder-decoder nature of the architecture as well as the bilingual format of the data. Let $N_e$ and $N_d$ be the number of non-embedding parameters in the encoder and the decoder respectively. Then, our proposed scaling law has the form

$$\hat{L}(N_e, N_d) = \alpha N_e^{-p_e} N_d^{-p_d} + L_\infty \tag{1}$$

where $\{\alpha, p_e, p_d, L_\infty\}$ are test set specific (fitted) parameters. In this formulation, $\alpha$ corresponds to the maximum loss reduction that one can hope from scaling, while $p_e$ and $p_d$ are the scaling exponents for encoder and decoder respectively. $L_\infty$ corresponds to the irreducible loss of the data.

In Figure 2, we fit our scaling law to the empirical loss values achieved by encoder/decoder scaling models on Web-Domain test sets. The results suggest that our scaling law almost fully captures the variation in the data ($R^2 > 99\%$). In particular, in contrast to traditional scaling laws, our proposed formulation is able to capture the different scaling behaviors of encoder and decoder scaled models.

While these improved fits are encouraging, to truly show that a candidate scaling law captures the model scaling behavior in NMT, it is necessary to verify the out-of-sample prediction power of the law. We provide two such verifications; we compare the predictions of the scaling law fitted on encoder / decoder scaling models with the empirical loss values from (i) symmetrically scaled models, (ii) models randomly sampled from the 2D grid of all potential encoder-decoder configurations.[5] Symmetrically scaled models are chosen for this verification as they are widely used by the NLP community (Raffel et al., 2020). Randomly sampled models are chosen to ensure that our results are not biased towards any particular model class.

Figures 3 & 4 present the out-of-sample results. As the plots suggest, the predictions of the scaling law match the empirical results with remarkable accuracy. These results suggest that Eq. (1) is able to capture the model scaling behavior across different scaling approaches and model sizes.

To ensure that our results generalize across different language pairs, we examine the fit of our scaling law on encoder / decoder scaling models trained on German→English (De→En), Chinese-to-English (Zh→En), and English-to-Chinese (En→Zh) translation tasks. Figure 5 presents the results evaluated on in-domain held-out data. Similar to the En→De case, our proposed functional form is able to closely capture the scaling behavior of these models.

To keep the scope of our experiments manageable, we focus on depth scaling in our analysis. In Appendix H, we ran large-scale experiments to see if other scaling approaches show significant deviations from the trends identified here. For large-scale models, which are the focus of this study, we observe that different scaling approaches yield very similar results: the final achieved test loss across different scaling approaches shows just $1-2\%$ variation. We observed that for small models ($< 100M$ parameters), this variation tends to grow, suggesting that model shape plays a key role in the performance of small models. We leave further analysis of this phenomenon to future research.

---

[5]See Appendix F for the full description of the models.

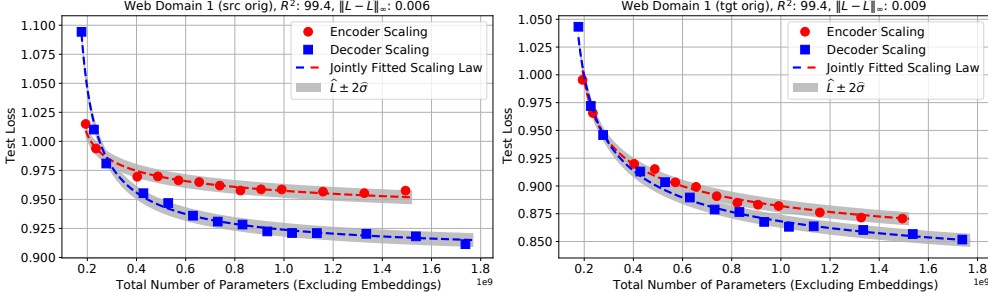

Figure 2: Evolution of test loss as a function of the model size for English→German models. Eq. (1) is jointly fitted to the empirical loss values from encoder scaling and decoder scaling experiments. Our proposed scaling law is able to capture more than $99\%$ of the variation in the data. We anticipate some fluctuations around the predicted trend ($\hat{\sigma} = 0.003$) caused by the randomness in the training pipeline (see Appendix E). See Appendix D for similar results on our other test sets.

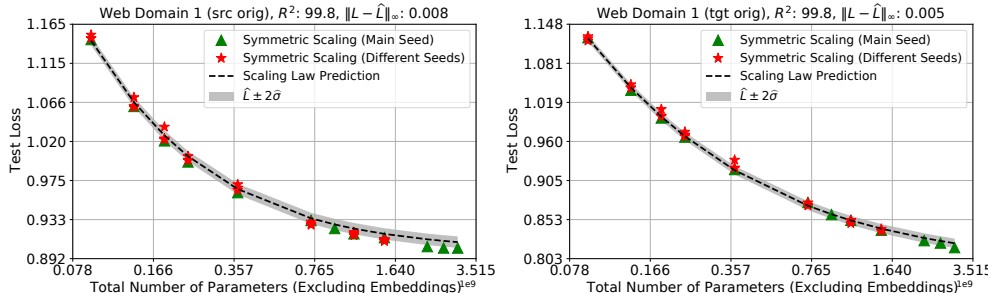

Figure 3: Comparison of the out-of-sample predictions of the scaling law with the empirical test loss values from symmetric scaling English→German models. Eq. (1) is fitted only using the encoder / decoder scaling data and then just evaluated on $(N_e, N_d)$ pairs corresponding to the symmetric scaling model. Our scaling law is able to almost fully capture the variation in the data ($R^2 = 99.8\%$). To examine the randomness in the results, we have repeated a subset of training runs with 4 different random seeds (see Appendix E for details). See Appendix D for similar results on our other test sets.

## 2.3 ANALYSIS

The above results suggest that the scaling law formalized in Eq. (1) captures the scaling behavior of the Transformer NMT models in multiple language pairs. As such, we can study the fitted coefficients to fully understand the scaling properties of these models. Several observations are in order:

**Decoder vs Encoder Scaling:** On all our test sets, the decoder exponents were observed to be larger than the encoder exponents, $p_d > p_e$ (see Figure 9). As a result, when improving the test loss is concerned, it is much more effective to scale the decoder rather than the encoder. This is contrary to the usual practice; due to latency considerations, many practitioners train NMT models with deep encoders and shallow decoders (Kasai et al., 2021). Our results suggest this practice could be sub-optimal in terms of loss reduction. Proposition 1 below provides guidance on how to allocate parameters in between the encoder and decoder optimally. The proof is presented in Appendix G.

**Proposition 1** (Optimal Scaling). *Assume the loss performance of the model is described by Eq. (1). Let B denote the budget for the total number of parameters. Then, the optimal encoder / decoder sizes (denoted respectively by $N_e^*$ and $N_d^*$) are $N_e^* = \frac{p_e}{p_e + p_d}B$ and $N_d^* = \frac{p_d}{p_e + p_d}B$. In addition, when optimally scaling the model, the scaling law reduces to $\hat{L}_{opt}(B) = \alpha^* B^{-(p_d + p_e)} + L_\infty$, where $\alpha^* \equiv \alpha\left(\frac{p_e + p_d}{p_e}\right)^{p_e}\left(\frac{p_e + p_d}{p_d}\right)^{p_d}$.*

Proposition 1 suggests that when $\frac{N_e}{N_d} = \frac{p_e}{p_d}$, the optimal scaling behavior can be achieved. Inspection of the functional form of Eq. (1) suggests that as long as $N_d/N_e$ is fixed as the model scales (i.e. encoder and decoder grow proportionally together), the optimal scaling exponent, $(p_e + p_d)$, can

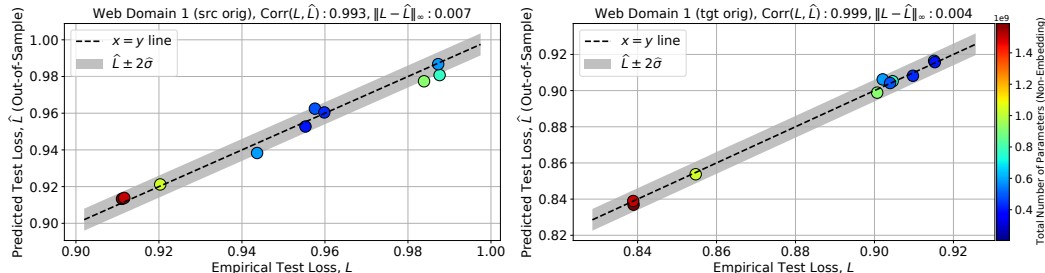

Figure 4: Comparison of the out-of-sample predictions of the scaling law with the empirical test loss values from the randomly sampled English→German models. The predictions are more than 99% correlated with the empirical test loss values. See Appendix F for similar results on other test sets.

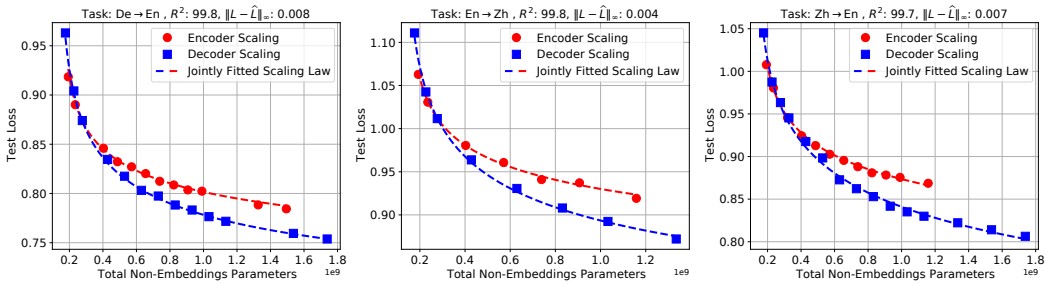

Figure 5: Fitted scaling laws for other language pairs. Eq. (1) is jointly fitted to the empirical loss value from encoder/decoder scaling experiments. Similar to En→De case, the law is able to describe the empirical scaling behavior of the models with high accuracy. See the appendix for more details.

be achieved, albeit with a potentially sub-optimal multiplicative constant, $\alpha^{\#}$. To examine the magnitude of this sub-optimality, in Figure 10, we compare the multiplicative constants resulting from proportional scaling of the encoder and decoder with different values of $N_d/N_e$. The results suggest that as long as the parameter allocation is not extremely far from $(N_e^*, N_d^*)$, the scaling scheme is approximately optimal. In particular, symmetrically scaling the encoder and decoder layers, is barely distinguishable from the optimal scaling scheme.

# 3 EFFECT OF DATASET COMPOSITION BIAS ON SCALING BEHAVIOR

Translation deals with the problem of mapping a sequence in one language into another language. A good translation should not only be adequate and fluent, but should ideally also adopt the style of a sentence naturally written in the target language. This necessitates MT models to make sense of natural looking inputs and generate natural looking outputs. As mentioned in Section 2, the examples used to train or test NMT models carry a critical bias, which we refer to as *composition bias*. Composition bias is introduced because of the unavailability of source-target examples (pairs) that are both natural in the accessible data generating distribution. For any given naturally generated text in a language, the corresponding text in the other language is either translated by humans, introducing *translationese* bias or translated by other machine translation systems, introducing *MT* bias. We consider both biases affecting the problem from a similar angle, hence we bundle them and call it composition bias. While machine translation by design has composition bias in the training/test sets employed (Freitag et al., 2020; Riley et al., 2020), its effect on model scaling is unknown. In this section we investigate the role of composition bias in scaling and identify critical factors playing a role.

We caution the reader to not take the composition bias as a problem specific to NMT. In fact as most training corpora in NMT are web-crawled, they can contain machine translation output on either the source or target side. Considering the growth of generated content in the web by machine learning

models [6] [7], it is not improbable that a proportion of the content collected and used by machine learning models is going to be biased by other models that are continuously generating content.

**The Effect of Test Set Construction:** We first examine the impact of composition bias on the test sets. Figure 9 shows the fitted scaling exponents for all our test sets. The plot suggests that the scaling powers for source-original test sets are drastically different from those of target-original test sets. This behavior is in direct contrast with language modeling setting (Kaplan et al., 2020) where it was observed that the evaluation on different test sets merely acted as a scaling penalty that only changed the multiplicative constants of the scaling law.

To elucidate this phenomenon further, in Figure 11 of the appendix, we compare the scaling trends for different source and target original test sets. Several observations are in order: Test sets with a similar composition approach (source or target original) have a qualitatively similar scaling behavior. However, scaling behavior is vastly different between the two composition approaches. Reducible loss quickly decays to zero for source original test sets. In fact, we observe that scaling our baseline 6L-6L model by a factor of 2.5 is sufficient for ensuring that reducible loss is below 0.05 for all source original test sets. In contrast, on target original test sets, the loss decays much more slowly with model size. For comparison, to ensure that reducible loss is below 0.05 for all target original test sets, we estimate that the baseline model has to be scaled up by a factor of 11.

Because of this behavior, the value of larger models in NMT is closely tied to their evaluation sets: On source original data, due to larger scaling exponents, even moderate increases in model size are sufficient for pushing the reducible loss close to zero. Hence, beyond a few hundred million parameters, there is no benefit in increasing the model size. In contrast, for target original data, which generally have smaller scaling exponents, large models are needed to push the reducible loss to zero.

**The Effect of Training Set Construction:** In this section, we briefly examine the role of *training data* construction on the scaling behavior. To do this, we generate two En→De datasets, that were not used in the previous experiments. One fully target original and another completely source original.

To generate the target original dataset, we compile a set of German documents from the web. Documents are screened to ensure the data is not machine generated. We use a Hybrid model (with 380M parameters) (Chen et al., 2018) to back-translate (BT) these documents to English. Similarly, for the source original data, we collect human generated English documents and (forward) translate them to German using a hybrid model (with approximately 327M parameters). Both datasets provide us with approximately 2.2 billion training examples. We mimic the experimental setup of Section 2.

Note that even though these datasets are not human generated, they reflect important aspects of training large NMT models. Many modern NMT datasets are harvested from the web and as a result, are contaminated with machine generated data. Moreover, many popular data augmentation algorithms such as Back Translation (Sennrich et al., 2016), sequence level distillation (Kim & Rush, 2016) and self training (He et al., 2020) purposefully add machine generated data into the training pipeline in order to take advantage of monolingual data.

Figure 6 describes the scaling behavior for models trained on target-original data. We observe that even though larger models are successful in reducing the training loss, they are unable to improve the test loss after roughly 400M parameters. Once this size threshold is exceeded, models overfit the training data and the test loss starts to deteriorate across all of our test sets. We hypothesize that this size threshold corresponds to the capacity of the original back-translation model. This assertion suggests that in order for back-translation to be beneficial for training large models, it has to be performed with a models with comparable capacity or higher. Although quite intriguing, we leave the verification of this hypothesis to future work.

Figure 7 paints another interesting picture for the models trained on the source-original data only, implying the target side having the composition bias, expected to be simpler, dull and not rich in its content, in short - not natural looking. As experiments suggest, even our smallest models are able to achieve extremely low loss values (roughly 0.16), with an apparent overfitting pattern. We believe the same phenomenon is also related to the "data simplification" effect sought by non-autoregressive models in NMT (Zhou et al., 2021).

---

[6]https://openai.com/blog/gpt-3-apps/
[7]https://blog.google/products/translate/one-billion-installs/

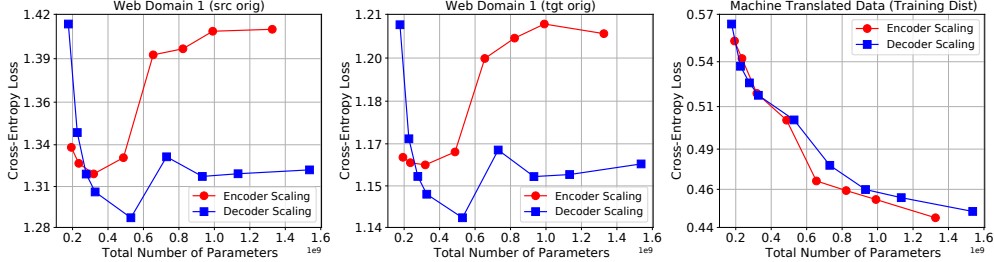

Figure 6: Scaling behavior of models trained on back-translated data. Right: Increasing the model size successfully reduces the loss on the training distribution. However, on the test data (left and center) the loss increases after approximately $400M$ parameters.

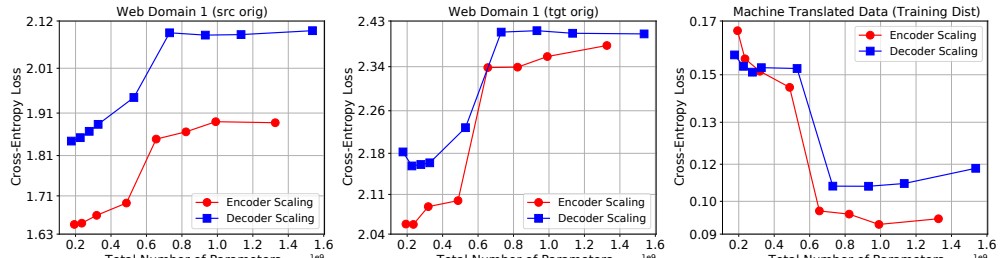

Figure 7: Scaling behavior of models trained on forward translated data. Left / center: early stopping test loss on Web-Domain. Right: loss at the end of the training for a subset of the training data.

## 4 EVOLUTION OF GENERATION QUALITY

We examine the effects of scaling on the output quality as measured by BLEU score [8]. For this analysis, we focus on output generated via beam search (Wu et al., 2016). For tractability purposes, we do not attempt to tune the (many) hyper-parameters of beam-search for each model. Instead, we use the configuration optimized for the baseline model (listed in Appendix J) in all the decoding tasks.

Figure 8 presents the co-evolution of BLEU score and cross-entropy loss throughout the training for all of our models. Depending on the construction of the test sets, two different empirical behaviors emerge. On target-original test sets, larger models are able to improve (lower) the test loss. These improvements in the loss are accompanied with consistent improvements (increases) in BLEU score. In fact, we observe that a simple power law can capture the relationship between BLEU score and cross-entropy loss for high-quality models. [9] In contrast, on source-original test sets, this relationship is absent; larger models consistently achieve better test losses, however, beyond a certain threshold, BLEU scores begin to deteriorate. Figures 27 and 28 exhibit that this phenomenon is not due to over-training; the BLEU score gap between large and small models is persistent throughout training.

To ensure that this observation truly reflects the generation quality of the models (as opposed to potential biases of BLEU score), we repeat our analysis with BLEURT score (Sellam et al., 2020a;b). The results are presented in Appendix K. The results show that BLEURT scores closely mirror the behavior of BLEU scores with respect to model scaling.

A careful look at the left-subplots of Figures 8 brings up another interesting trend. At similar values of the test loss, encoder-scaled models result in better generation quality compared to decoder-scaled models. This findings agrees with previous work that relied on encoder-scaling when optimizing for BLEU and inference latency (Kasai et al., 2021). Whether these differences in the effects of encoder-scaling and decoder-scaling are caused by insufficient search algorithms, or just different model fits from different architectural priors is left to future work.

---

[8]We computed the BLEU scores using an internal reimplementation of Moses scorer: `mteval-v13a.pl`.

[9]Fitting details are presented in Appendix J.

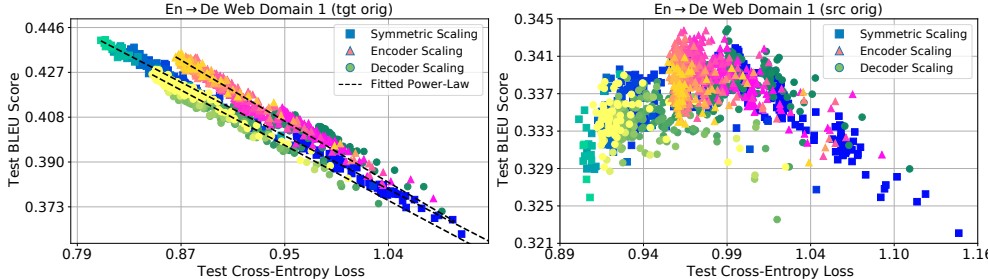

Figure 8: Log-log plot of the evolution of BLEU score as a function of test loss. For each scaling approach, warmer colors represent larger models. Each individual color represents different check-points of a single model during training. On target original data, improvements to cross-entropy loss lead to consistent improvements in BLEU score. Dashed lines correspond to power-law fits. The relationship breaks down for source original data. See Appendix J for more details.

## 5 CONCLUSION AND LIMITATION

In this work we have attempted to quantify the evolution of model quality as a function of model capacity for encoder-decoder NMT models. While a univariate scaling law describing the cross-entropy as a function of the total number of parameters in the model is insufficient, a bivariate law treating the number of encoder and decoder parameters as separate variables adequately describes the scaling behavior of these models under various scaling strategies. We validate this behavior on a variety of language pairs and evaluation sets. Whether this behavior is intrinsic to the encoder-decoder architecture, or arising from the nature of the NMT task, requires further study.

Next, we show that this scaling behavior is highly dependent on the composition of the evaluation data, specifically on whether the source or target sentences are "original". Our findings indicate that target-original evaluation sets continue benefiting from model scaling, while the reducible error on source-original evaluation sets quickly saturates to $0$. This could be an artifact of the lack of diversity in translated text; a simpler target distribution doesn't require much capacity to model while generating fluent or natural-looking text could benefit much more from scale.

We also study how the composition of training data affects the scaling behavior of models. When training on target-original (back-translated) text, model quality keeps improving until a point after which the trend saturates. In our study the capacity where saturation manifests first is perilously close to the capacity of the model used for back-translation, indicating that the capacity of the generative model used to generate synthetic text might have a role to play, but this requires further investigation. When training on source-original text, even low-capacity models are sufficient to reach the irreducible loss region, painting a gloomy picture for synthetic data. While we have explored these ideas in the context of machine translation, given the proliferation of generative models this problem will likely be a challenge for future practitioners training on web-scraped monolingual datasets as well. For low-resource languages, the proliferation of machine translated text is already a problem given that a significant portion of web text in these languages is machine translated.

Finally, we attempt to understand how generation quality evolves with the improvements in cross-entropy resulting from model scaling. As with our previous findings, dataset composition plays a major role in determining the trends. For source-original evaluation sets, the correlation between cross-entropy and generation quality breaks down. On target-original evaluation, we observe an inverse correlation between cross-entropy and BLEU, suggesting that improved model fit results in a corresponding improvement in generation quality. The slope of this relationship is different for encoder-scaling and decoder-scaling, with encoder-scaled models performing better on BLEU than decoder-scaled models, at the same level of cross-entropy loss. Whether this is an artifact of our search strategy or the difference in architectural priors is something that requires further investigation.

Our findings suggest that scaling behavior of encoder-decoder NMT models is predictable, but the exact formulation of scaling laws might vary depending on the particular architecture or task being studied. Our empirical findings also raise concerns regarding the effect of synthetic data on model scaling and evaluation, and how proliferation of machine generated text might hamper the quality of future models trained on web-text.

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

## A  CONNECTIONS TO RECENT LITERATURE

In this section, we provide a more detailed overview of the scaling laws literature immediately relevant to our investigation. Hestness et al. (2017) examine data scaling laws for LSTM-based encoder-decoder NMT models. They show that the model performance exhibits an empirical power-law relationship with the size of the training data. Rosenfeld et al. (2019) study model and data scaling laws for image classification and language modeling tasks. They provide a bivariate scaling law that jointly capture the model and data scaling behaviors for each of these tasks. Crucially, they show that scaling laws can be used to effectively extrapolate beyond model/data configurations used for fitting them. In particular, they show that one can use smaller models to fit the scaling law and then use the law to accurately predict the performance of larger models. Our results (Figures 3 and 4) validate that this behavior also holds for our setting.

Kaplan et al. (2020) study the scaling behavior of decoder-only Transformer models used in language modeling. They identify three factors that principally determine the model performance: the number of (non-embedding) model parameters, the size of the training dataset, and the amount of compute used for training. In the scenario where data and compute are abundant, they show that model performance exhibits an empirical power-law relationship with the total number of (non-embedding) parameters in the network. Henighan et al. (2020) verify this power-law behavior for other auto-regressive generative modeling tasks such as image modeling, video modeling, and mathematical problem solving. In contrast, in our setting, we observe that the total number of parameters is not a good predictor of the test loss in NMT. Instead, we show that encoder-decoder NMT models scale according to a bivariate function of the number of encoder parameters and the number of decoder parameters.

Kaplan et al. (2020) also examine how scaling affects the out-of-distribution generalization behavior in language models. Their results suggest that different test sets exhibit rather similar scaling trends as model size increases. In comparison, in the NMT setting, we show that different test sets, depending on their naturalness structure, exhibit drastically different scaling behaviors.

The most closely related study to ours is the work of Gordon et al. (2021). They also examine scaling laws for NMT, albeit at a different parameter regime: Gordon et al. focus on small models (393K-56M parameters) while here, we focus on large scale models (100M-4B parameters). In addition, we examine the general scenario where encoder and decoder can scale asymmetrically (which is a common practice in NMT) while Gordon et al. only consider symmetric scaling. Despite the differences in the experimental setup, when focusing on the symmetric scaling setting, the conclusions of both studies mostly align. In particular, when symmetrically scaling the model, our scaling law simplifies to

$$L(N) = \alpha N^{-p} + L_\infty$$

which agrees with the conclusions of Gordon et al. One difference in our conclusions is the presence of the $L_\infty$ term in our scaling law. This term, which captures the irreducible loss in the learning problem, does not appear in Gordon et al's analysis. We suspect that this might be due to the small models used in that analysis which cannot statistically observe such asymptotic terms.

Gordon et al. also examine the relationship between test cross-entropy loss and BLEU score. They observe that as models scale and cross-entropy loss improves, BLEU score also improves in tandem. In comparison, we unveil a more intricate relationship in our study: We show that when the test data is target-original, improvements in the test cross-entropy loss are correlated with improvements in generation quality (as measured by BLEU/BLEURT scores). However, when the test set is source-original, the correlation between cross-entropy loss and generation quality breaks down. In this setting, we observe that generation quality improves up until a certain threshold and then deteriorates (see the right plot in Figure 8).

## B    MISCELLANEOUS FIGURES

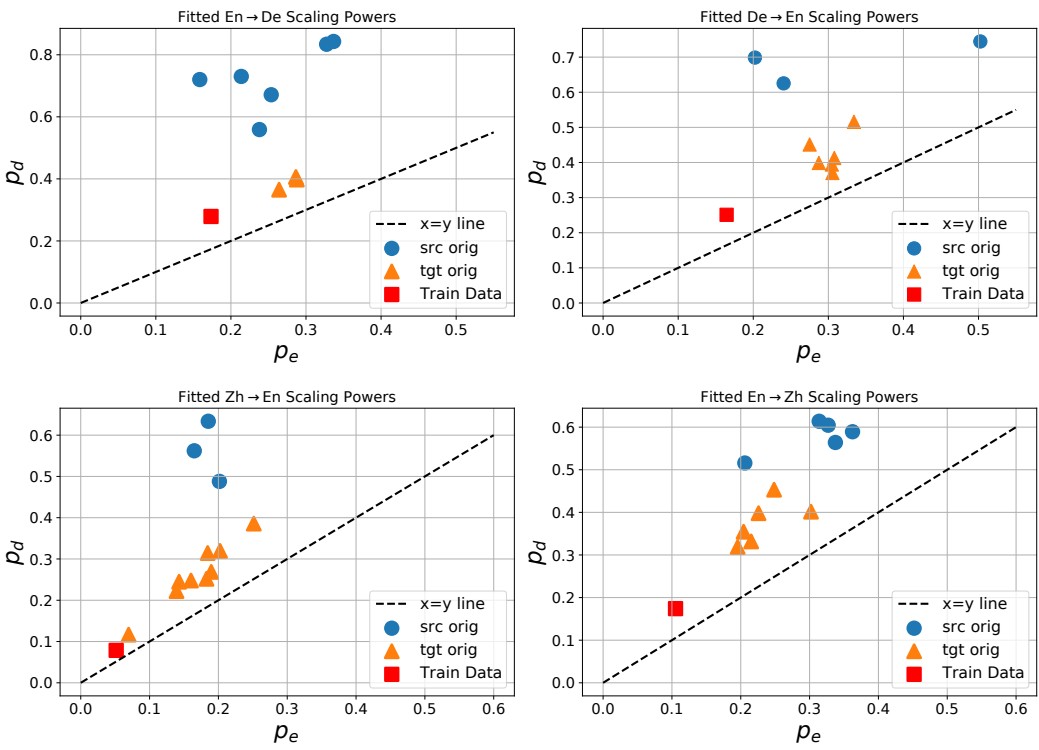

Figure 9: Fitted scaling exponents for En↔Zh and En↔De translation tasks. Across all the test sets under consideration, we observe $p_d > p_e$.

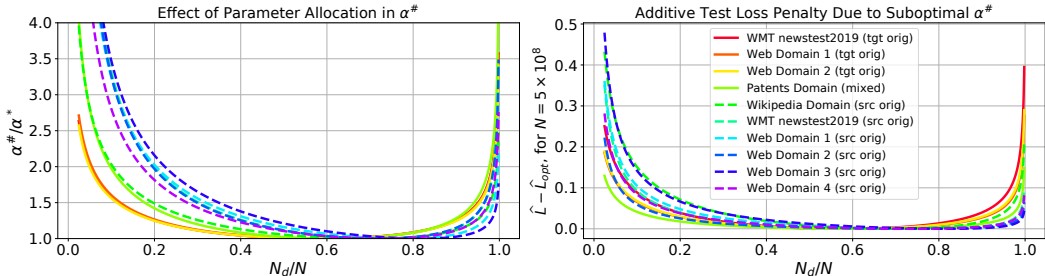

Figure 10: We use our fitted scaling laws to evaluate the effect of encoder / decoder parameter allocation ratio when proportionally scaling the encoder and the decoder. Left: $\alpha^{\#}/\alpha*$ for different parameter allocation schemes. Right: The predicted additive loss penalty, $(\widehat{L} - \widehat{L}_{opt})$, for a model with $5 \times 10^8$ total (non-embedding) parameters. Each line corresponds to a different test set.

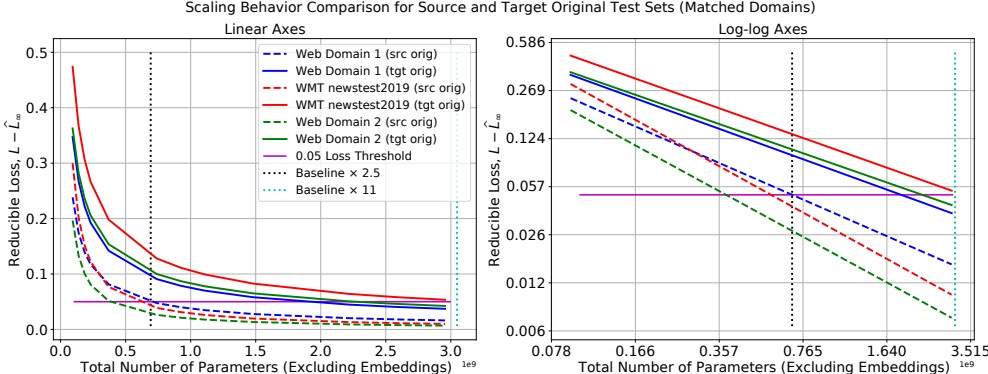

Figure 11: A comparison of scaling behavior across source and target original test sets. We use our fitted scaling laws to estimate the evolution of reducible loss for each test set. All scaling trends correspond to symmetrically scaling the encoder and decoder layers.

## C   ARCHITECTURE AND HYPER-PARAMETER DETAILS

As described in Section 2, all our models use a similar configuration for their Transformer Blocks. In particular, we fix model dimension to $1024$, feed-forward layer dimension to $8192$, number of attention heads to $16$, and attention hidden dimension to $1024$. Our models use a sentence-piece vocabulary of size $32000$.

**Regularization:**   We use a dropout of $0.1$ for residuals, feed-forward activations and attention. Models are trained with label smoothing of magnitude $0.1$. To improve the training stability, all models use logit clipping of $10$.

**Optimizer:**   We use Adafactor (Shazeer & Stern, 2018) optimizer for training our models. We use $40$k linear warm-up steps and an inverse square root learning rate schedule. For Adafactor we used momentum with $0.9$ and factored second moment to save memory.

Table 1 (and Table 2 resp.) describes the parameter decomposition of the encoder scaling (decoder scaling) models. Table 3 describes the parameter counts for the symmetric scaling models. The largest model we used (64L-64L) has more than 3 billion parameters while the smallest model we used (2L-2L) has only 92M non-embedding parameters.

Table 1: Parameter decomposition of the encoder scaling models. The total number of parameters includes 98M parameters representing the softmax and embedding layers.

| Encoder | | Decoder | | |
|---|---|---|---|---|
| Layers | Parameters | Layers | Parameters | Total Parameters |
| 2 | 42M | 6 | 151M | 291M |
| 4 | 84M | 6 | 151M | 333M |
| 8 | 168M | 6 | 151M | 417M |
| 12 | 252M | 6 | 151M | 501M |
| 16 | 336M | 6 | 151M | 585M |
| 20 | 420M | 6 | 151M | 669M |
| 24 | 504M | 6 | 151M | 753M |
| 28 | 588M | 6 | 151M | 837M |
| 32 | 672M | 6 | 151M | 921M |
| 36 | 756M | 6 | 151M | 1005M |
| 40 | 840M | 6 | 151M | 1089M |
| 48 | 1007M | 6 | 151M | 1257M |
| 56 | 1175M | 6 | 151M | 1425M |
| 64 | 1343M | 6 | 151M | 1593M |

Table 2: Parameter decomposition of the decoder scaling models. The total number of parameters includes 98M parameters representing the softmax and embedding layers. Note that the 6L-6L model is the baseline model we used for hyper-parameter tuning.

| Encoder | | Decoder | | |
|---|---|---|---|---|
| Layers | Parameters | Layers | Parameters | Total Parameters |
| 6 | 126M | 2 | 50M | 275M |
| 6 | 126M | 4 | 101M | 325M |
| 6 | 126M | 6 | 151M | 375M |
| 6 | 126M | 8 | 202M | 426M |
| 6 | 126M | 12 | 302M | 527M |
| 6 | 126M | 16 | 403M | 627M |
| 6 | 126M | 20 | 504M | 728M |
| 6 | 126M | 24 | 605M | 829M |
| 6 | 126M | 28 | 705M | 930M |
| 6 | 126M | 32 | 806M | 1030M |
| 6 | 126M | 36 | 907M | 1131M |
| 6 | 126M | 40 | 1008M | 1232M |
| 6 | 126M | 48 | 1209M | 1433M |
| 6 | 126M | 56 | 1411M | 1635M |
| 6 | 126M | 64 | 1612M | 1836M |

Table 3: Parameter decomposition of the symmetric scaling models trained for English→German translation task. The total number of parameters includes 98M parameters representing the softmax and embedding layers.

| Encoder | | Decoder | | |
|---|---|---|---|---|
| Layers | Parameters | Layers | Parameters | Total Parameters |
| 2 | 42M | 2 | 50M | 191M |
| 3 | 63M | 3 | 76M | 237M |
| 4 | 84M | 4 | 101M | 283M |
| 5 | 105M | 5 | 126M | 329M |
| 8 | 168M | 8 | 202M | 468M |
| 16 | 336M | 16 | 403M | 837M |
| 20 | 420M | 20 | 504M | 1022M |
| 24 | 504M | 24 | 605M | 1207M |
| 32 | 672M | 32 | 806M | 1576M |
| 48 | 1007M | 48 | 1209M | 2315M |
| 56 | 1175M | 56 | 1411M | 2684M |
| 64 | 1343M | 64 | 1612M | 3054M |

## D    SCALING LAWS FOR OTHER TEST SETS

In order to keep the discussion in the main text focused, we only presented scaling laws for Web Domain 1 test sets. These test sets were chosen as they had a domain similar to the training data (i.e. web). In this appendix, we repeat the same analysis for our other test sets. The details of these test sets are described in Section 2.

Figure 12 demonstrates how well the scaling law in Eq. (1) fits the empirical scaling behavior of our models on all our test sets. For each test set, we have fitted the law jointly on the final test loss achieved by the encoder and decoder scaling models. We measure the final test loss by the median test loss over steps 450K to 500K. Details of the fitting procedure are provided in Appendix I.

The results suggest that Eq. (1) is closely capturing the scaling behavior of the model for all the test sets / domains. In the last row, we also demonstrate the fit for the training data (left column) and the training loss (cross entropy on training data plus regularization, right column). We observe that the scaling law is almost perfectly fitting the empirical data in these cases.

To examine the fit more closely, in Figure 13, we have plotted the same data but with several modifications:

1. Instead of plotting the final loss, we plot the *reducible* component of the final loss ($L - L_\infty$). As the true value of $L_\infty$ is unknown, we use the value given by the fit of the scaling law.

2. For encoder scaling models, we plot the (reducible) loss against the number of encoder parameters (as opposed to the total number of parameters). Similarly, for decoder scaling models, we plot the loss against the number of decoder parameters.

3. We use log-log scaling on the axes.

4. We use the results of Table 4 to provide a confidence region around our predictions. This confidence region quantifies our expected uncertainty caused by randomness in the initialization and training pipeline.

Eq. (1) predicts that the relationship between the empirical final loss values from the encoder (decoder) scaling models and the number of encoder (decoder) parameters should appear linear on these plots. Figure 13 suggests that the empirical scaling behavior of these models conforms closely to these predictions.

To see if Eq. (1) continues to capture the scaling behavior of the models out-of-sample, we compare the predictions of the scaling law with the empirical loss values for symmetric scaling models. In other words, we examine how well the scaling laws fitted only using encoder / decoder scaling models predict the final test loss achieved by symmetric scaling models of different sizes. Figure 14 shows this comparison for all of our test sets. We observe a remarkable match between the predictions of the scaling law and the empirical loss values across the board. These observations confirm that Eq. (1) is able to capture the scaling behavior of the model regardless of the scaling approach.

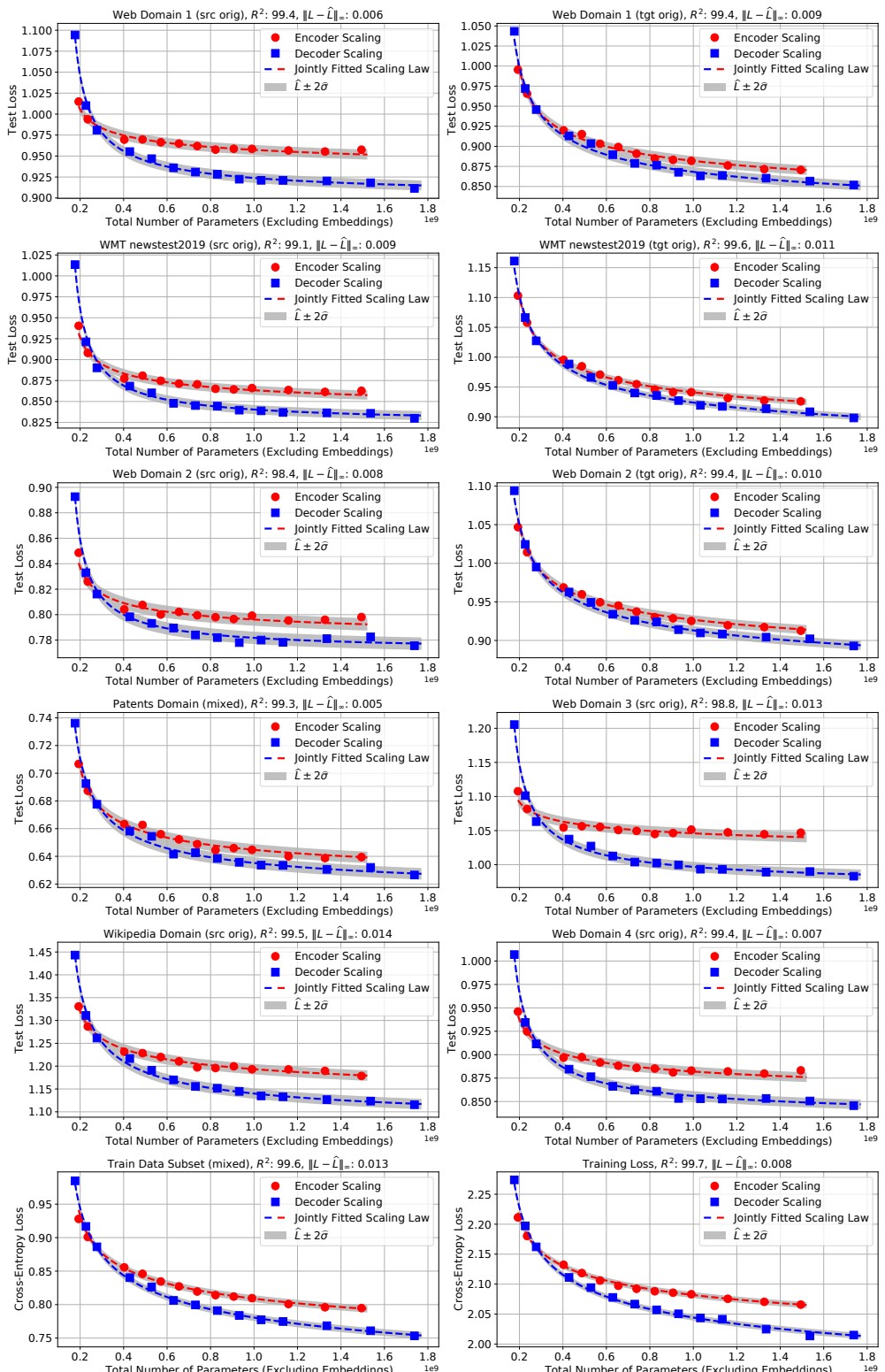

Figure 12: Fitted scaling law (Eq. (1)) for English→German translation task. The scaling law captures the scaling behavior of the models over a diverse collection of test sets and domains. The last row describes the evolution of the cross-entropy loss on the training data (with and without regularization effect).

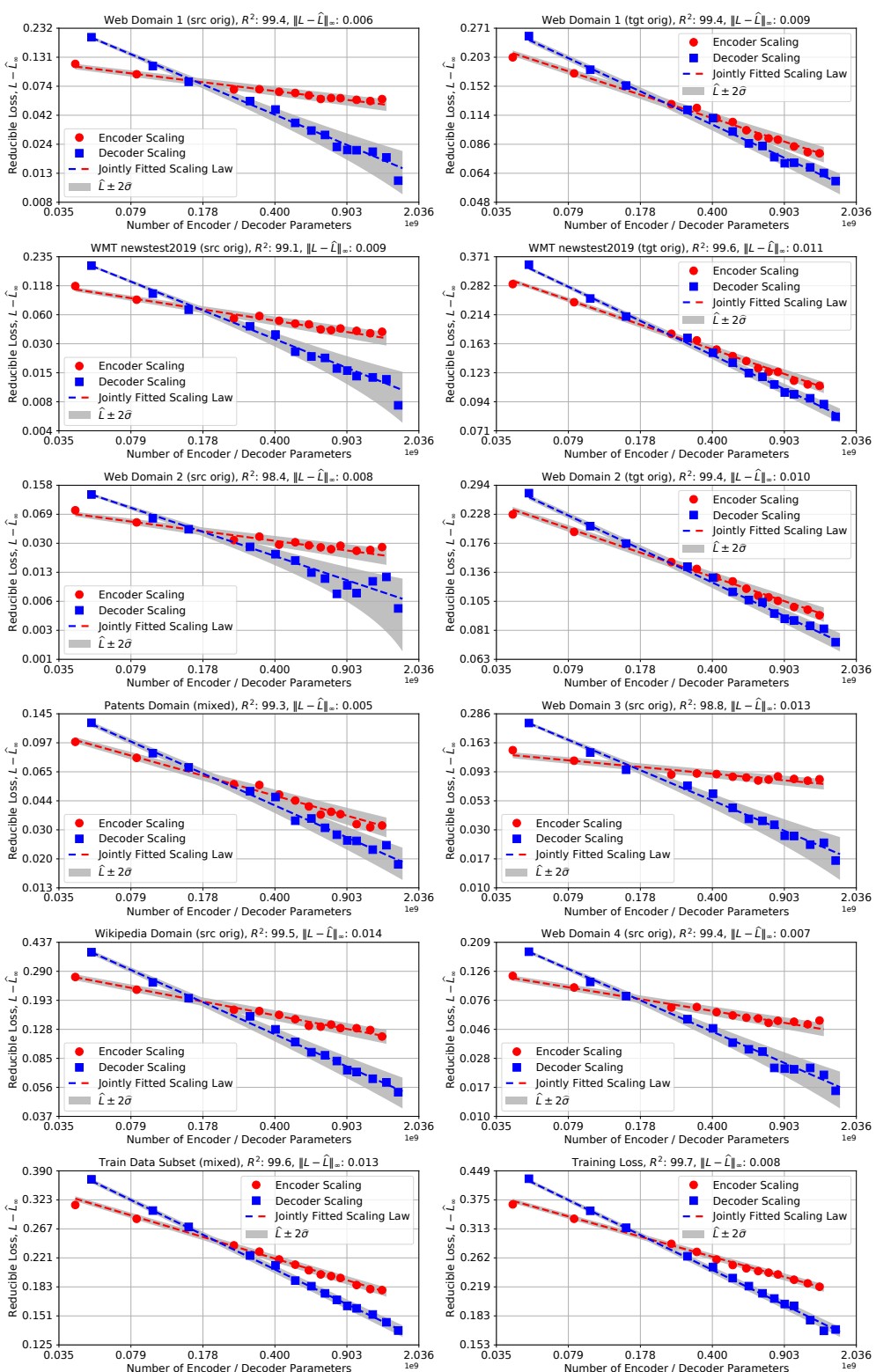

Figure 13: Fitted scaling law (Eq. (1)) for English→German translation task. Here, we use a log-log plot in order to inspect the fit more closely. Shaded cyan regions correspond to the uncertainty region given by ±2×standard deviation. Per test set standard deviations are provided in Table 4.

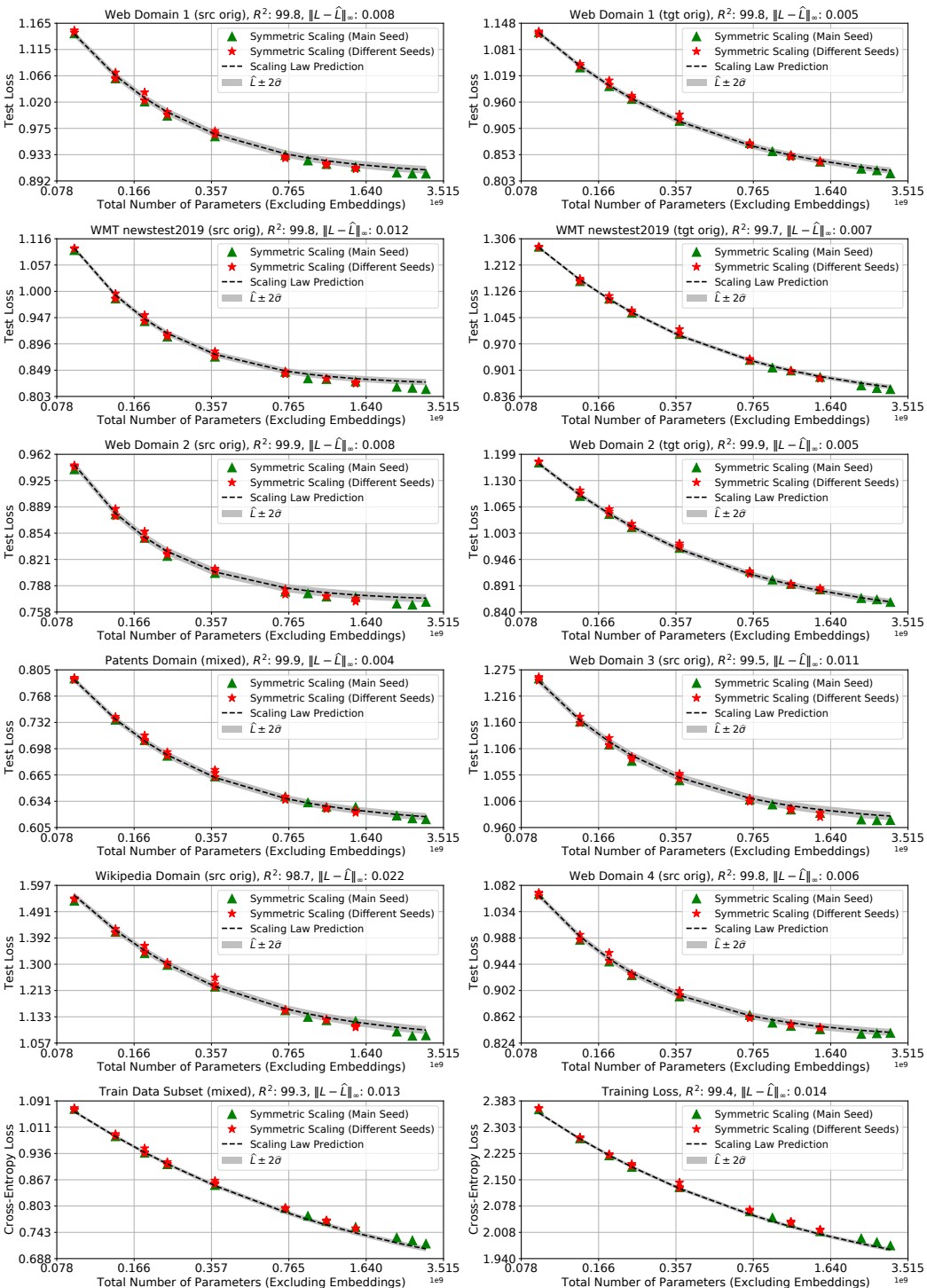

Figure 14: Out-of-sample prediction accuracy of English→German scaling laws on symmetric scaling models. Scaling laws are fitted only using the encoder and decoder scaling models. Nevertheless, they accurately predict the scaling behavior of symmetric scaling models.

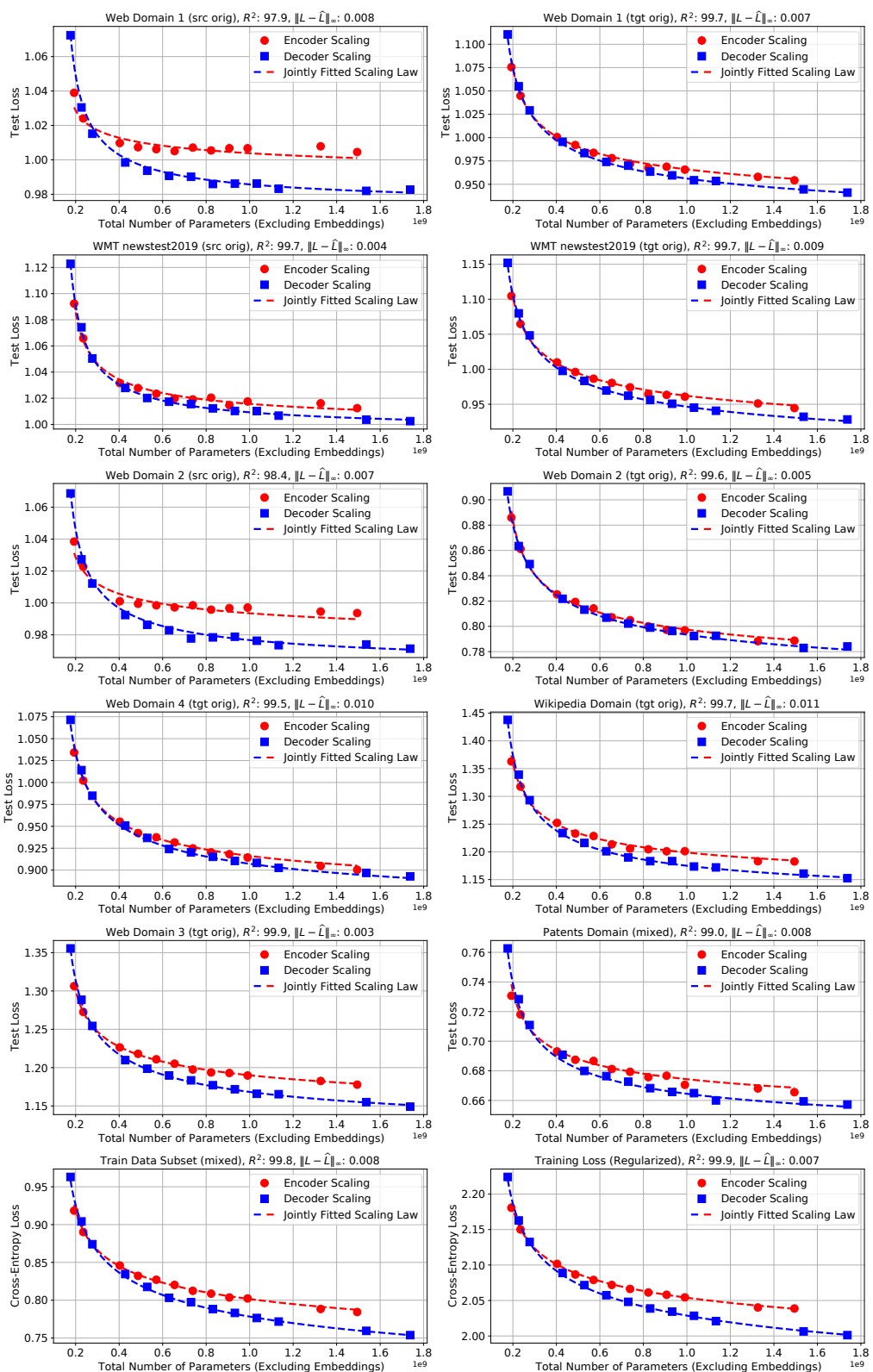

Figure 15: Fitted scaling law (Eq. (1)) for German→English translation task. The scaling law captures the scaling behavior of the models over a diverse collection of test sets and domains.

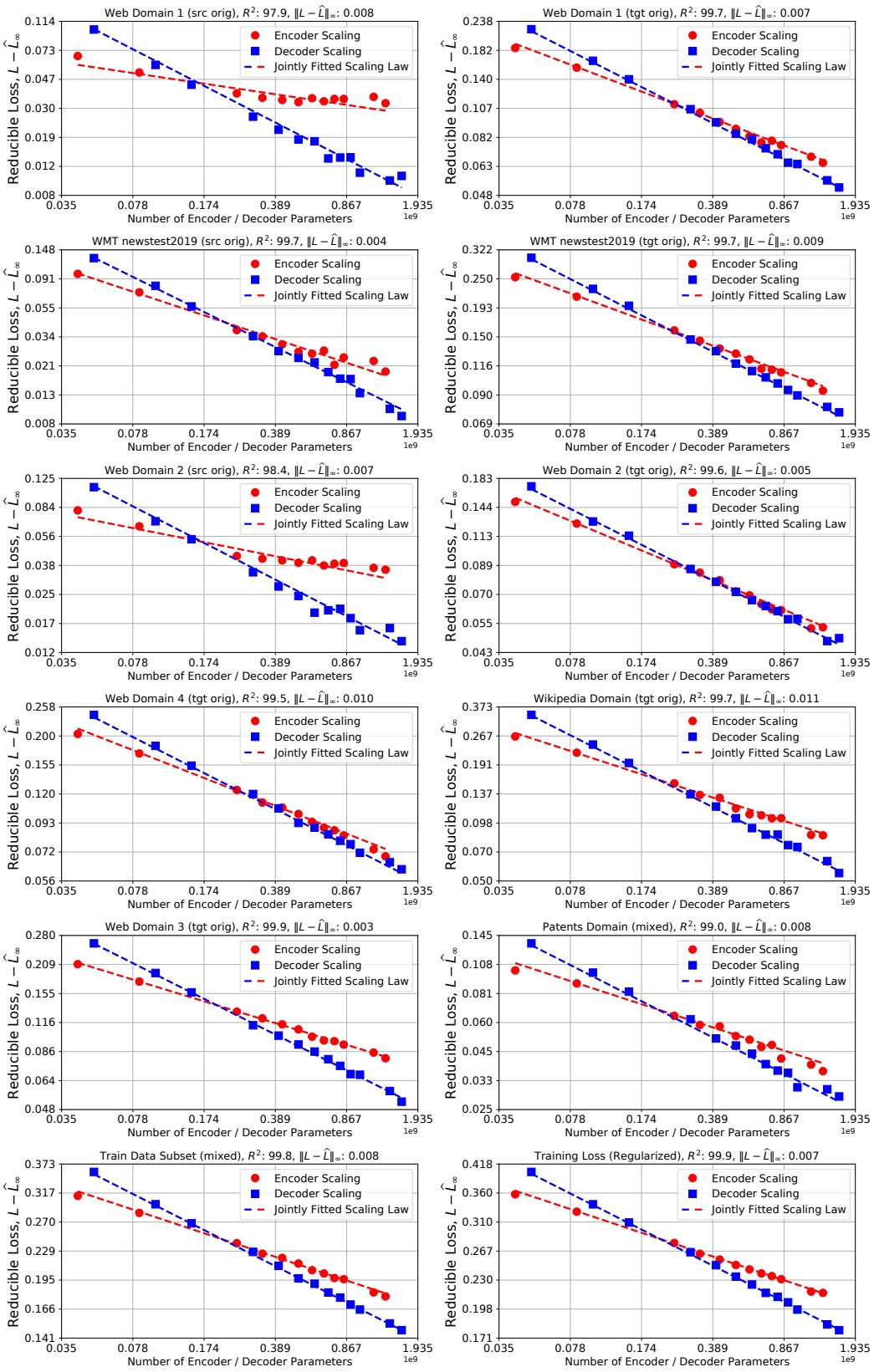

Figure 16: German→English scaling law fits on log-log scale.

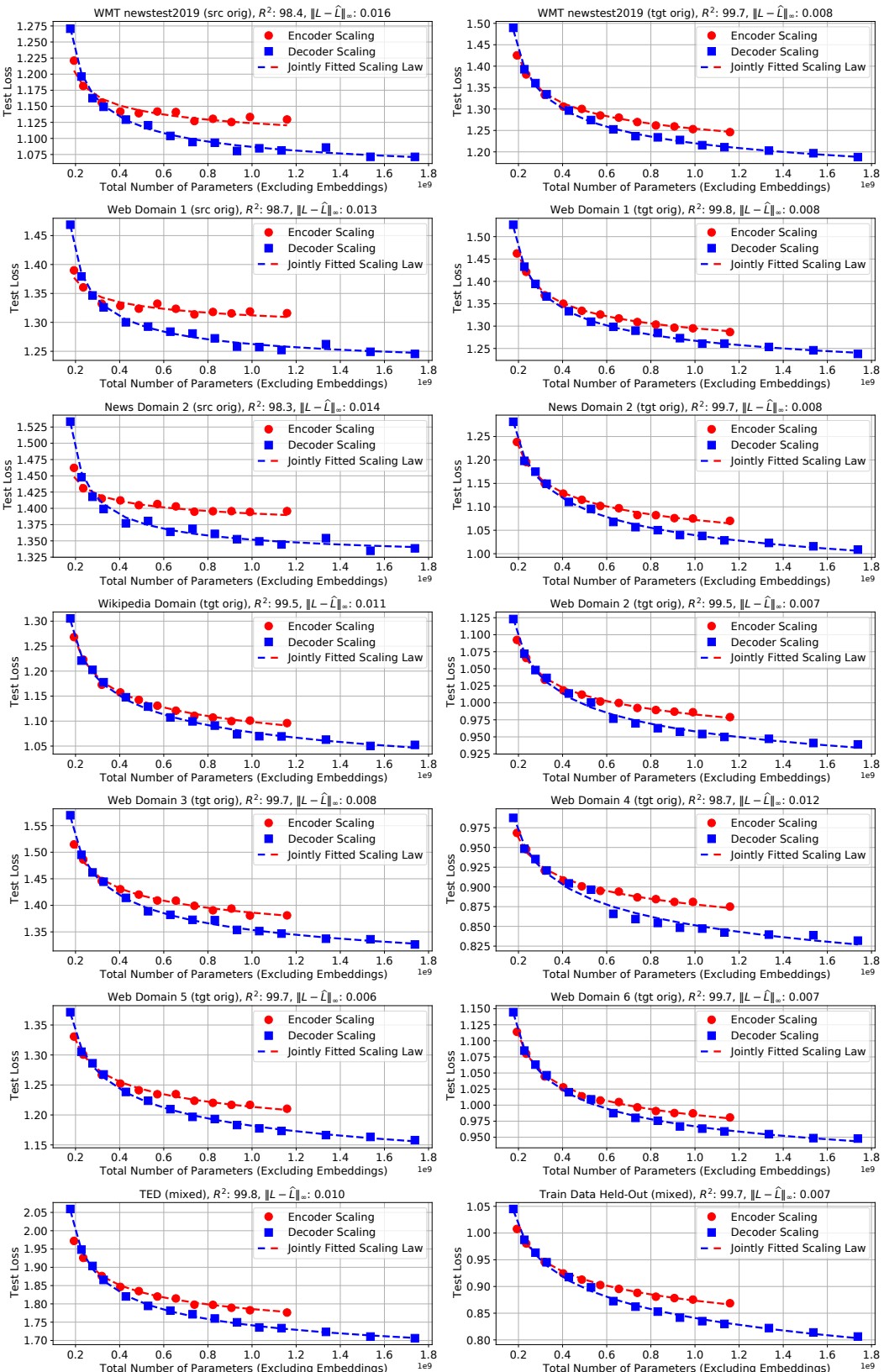

Figure 17: Chinese→English scaling law fits.

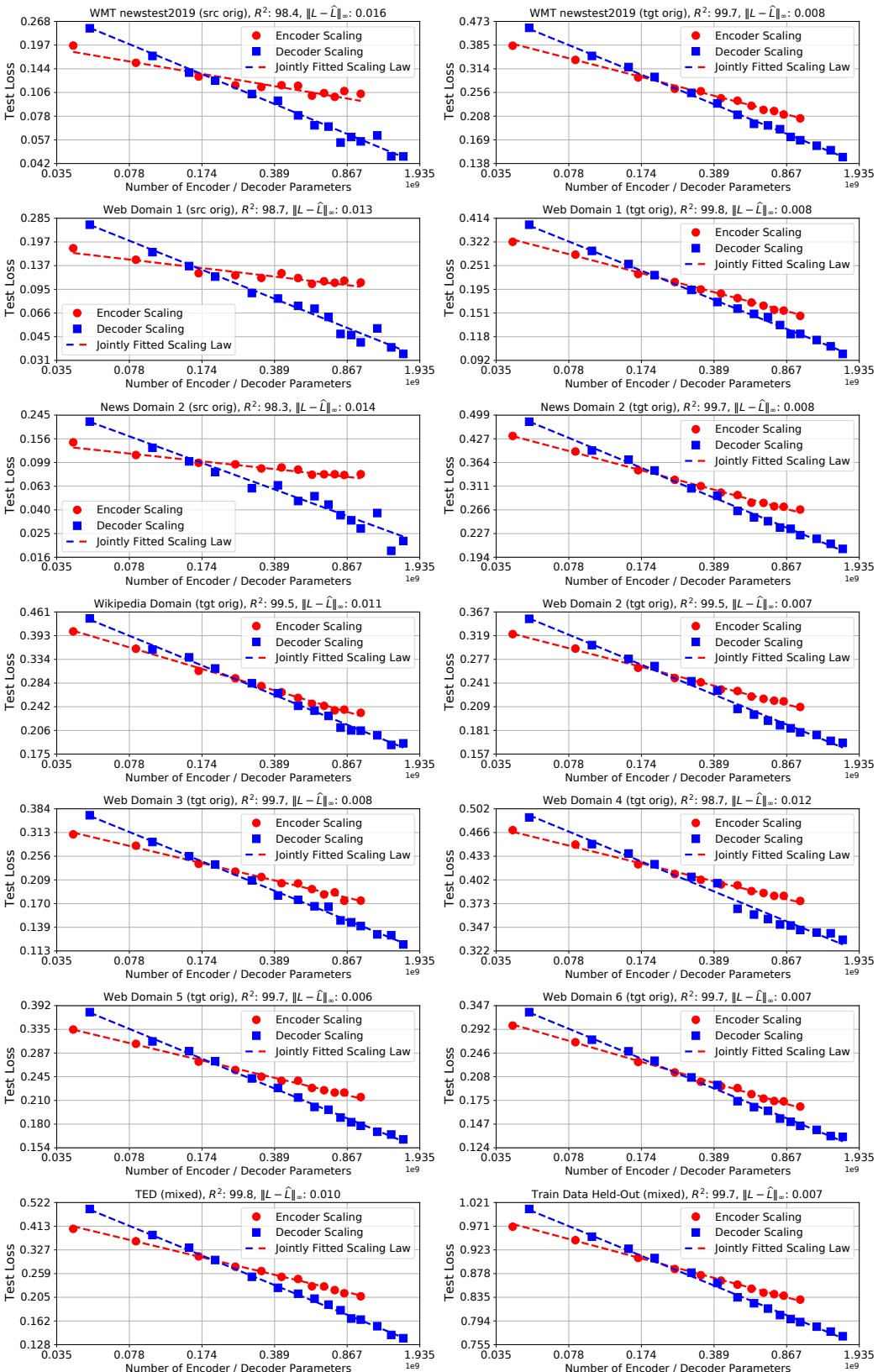

Figure 18: Chinese→English scaling law fits on log-log scale.

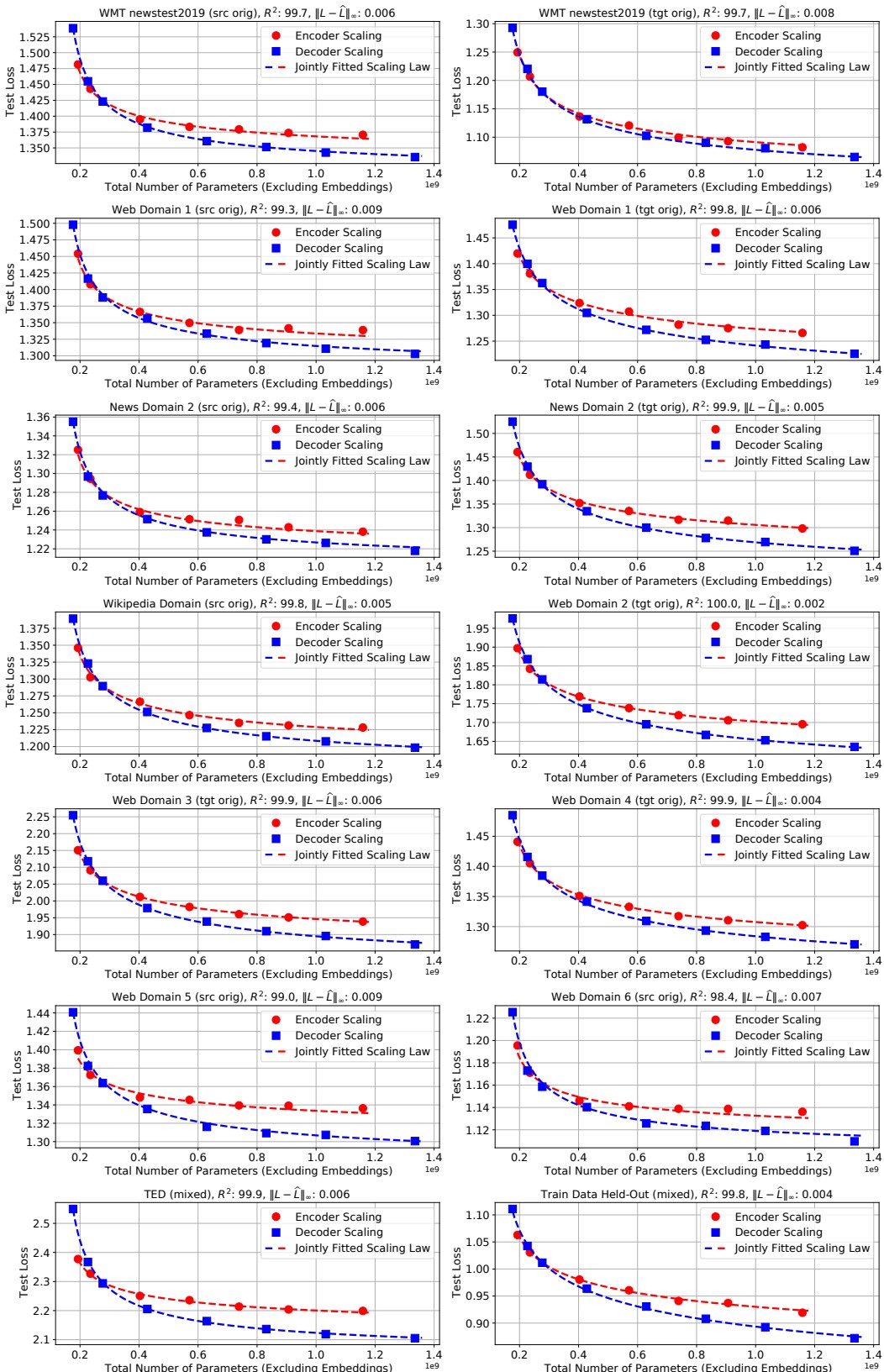

Figure 19: English→Chinese scaling law fits.

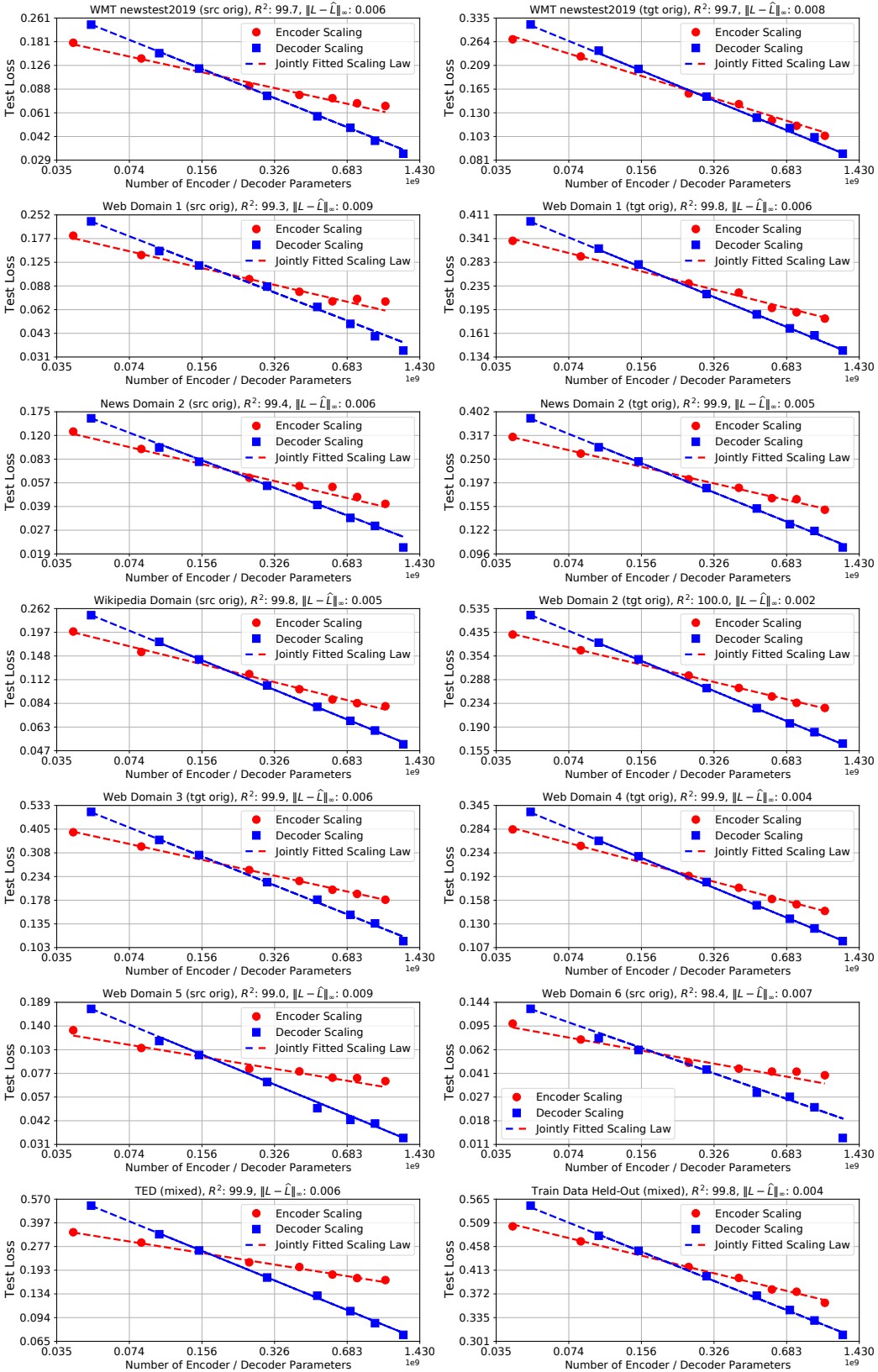

Figure 20: English→Chinese scaling law fits on log-log scale.

# E Quantifying the Random Variations in the Results

Note that the final test loss achieved by the model is a random quantity. Randomness is incorporated into the training pipeline through the initialization step, data order, and hardware failures / preemptions. To quantify the magnitude of the fluctuations caused by this randomness, we retrain a subset of our models (2L-2L, 3L-3L, 4L-4L, 5L-5L, 8L-8L, 16L-16L, 24L-24L, and 32L-32L) with 4 different seeds. Figure 21 presents standard deviation (left) and maximum difference (right) of the final test loss values observed for each model.

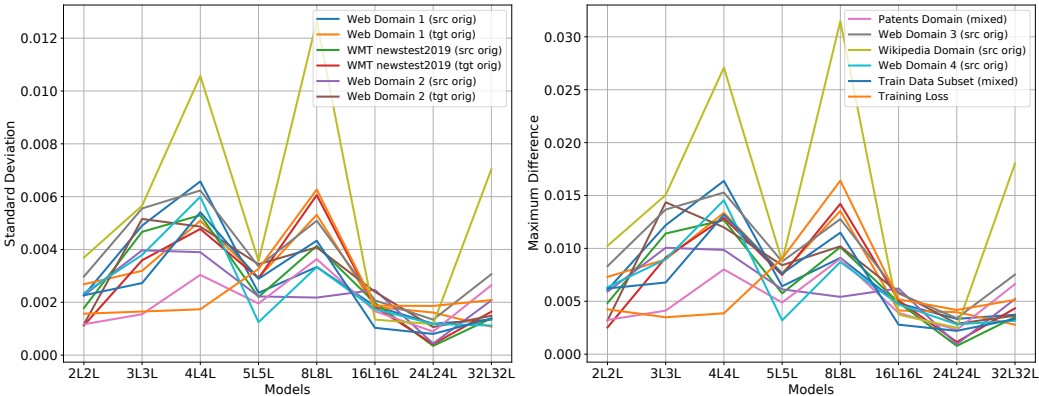

Figure 21: Variability of the final test loss across four different seeds.

Table 4: Variability of final test loss for each test dataset (averaged over all models).

| Dataset | Average Standard Deviation | Average Maximum Deviation |
|---|---|---|
| Web Domain 1 (src orig) | 0.0030 | 0.0078 |
| Web Domain 1 (tgt orig) | 0.0030 | 0.0077 |
| WMT newstest2019 (src orig) | 0.0027 | 0.0067 |
| WMT newstest2019 (tgt orig) | 0.0028 | 0.0071 |
| Web Domain 2 (src orig) | 0.0024 | 0.0062 |
| Web Domain 2 (tgt orig) | 0.0030 | 0.0076 |
| Patents Domain (mixed) | 0.0021 | 0.0053 |
| Web Domain 3 (src orig) | 0.0037 | 0.0094 |
| Wikipedia Domain (src orig) | 0.0057 | 0.0146 |
| Web Domain 4 (src orig) | 0.0026 | 0.0066 |
| Train Data Subset (mixed) | 0.0026 | 0.0067 |
| Training Loss | 0.0025 | 0.0064 |

# F  RANDOMLY SAMPLED MODELS

To ensure that our scaling laws are not biased towards a particular class of models, we test their out-of-sample predictions on 10 models where the number of encoder and decoder layers are decided randomly. We use the following code to generate randomly sampled models:

```
np.random.seed(99)
decoders = np.random.randint(3, 40, size=10)
encoders = np.random.randint(3, 40, size=10)
```

Table 5 lists the parameter allocation of these randomly sampled models.

Table 5: Parameter decomposition of the randomly sampled models trained for English→German translation task. The total number of parameters includes 98M parameters representing the softmax and embedding layers.

| Encoder | | Decoder | | |
|---|---|---|---|---|
| Layers | Parameters | Layers | Parameters | Total Parameters |
| 5 | 105M | 12 | 302M | 506M |
| 11 | 231M | 8 | 202M | 531M |
| 15 | 315M | 7 | 176M | 590M |
| 23 | 483M | 4 | 101M | 682M |
| 3 | 63M | 21 | 529M | 690M |
| 32 | 672M | 4 | 101M | 871M |
| 39 | 819M | 4 | 101M | 1018M |
| 18 | 378M | 26 | 655M | 1131M |
| 26 | 546M | 38 | 957M | 1601M |
| 30 | 630M | 38 | 957M | 1685M |

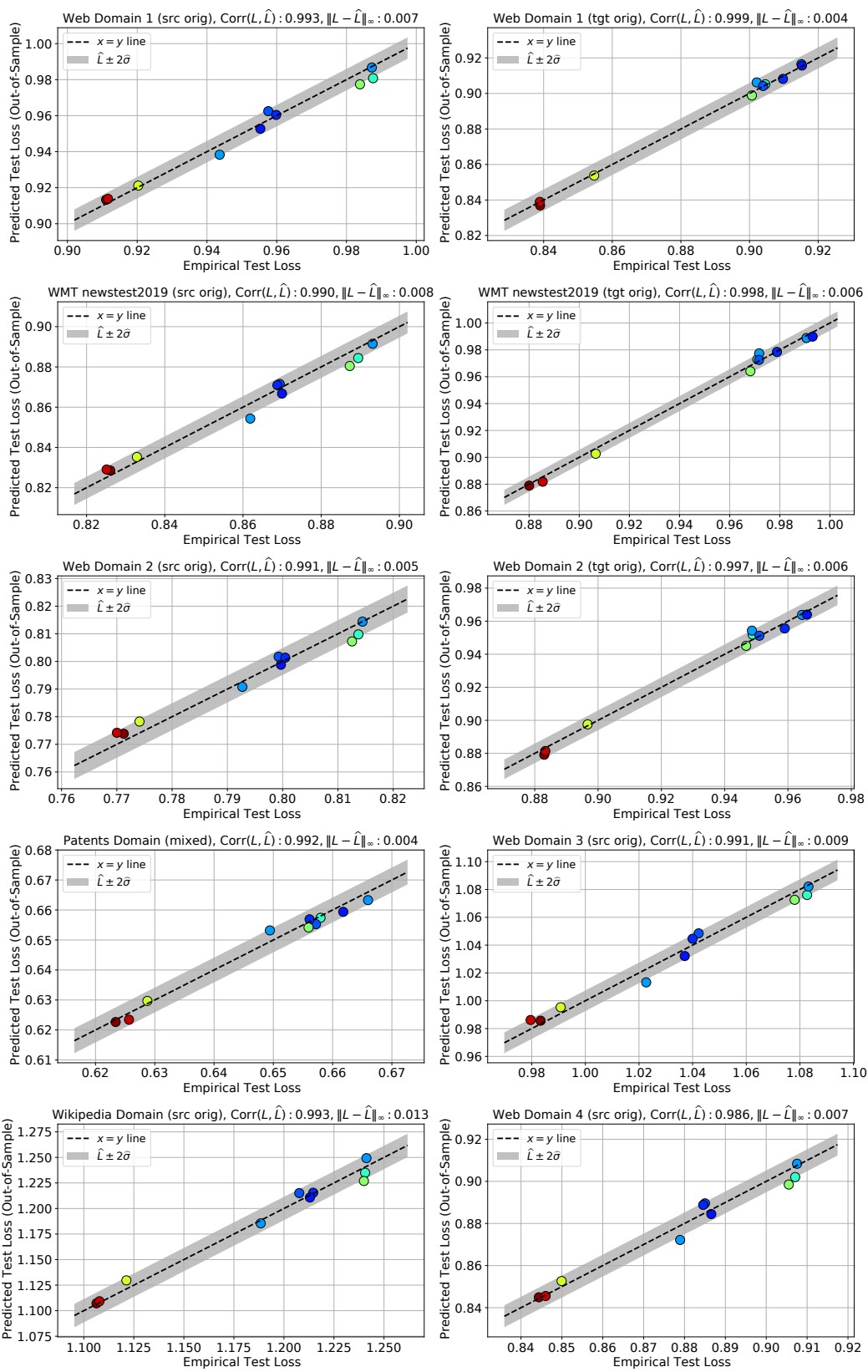

Figure 22: Comparison of the out-of-sample predictions of the scaling law with the empirical test loss values from the randomly sampled English→German models.

## G PROOFS

### G.1 PROOF OF PROPOSITION 1

*Proof.* The optimal encoder / decoder sizes are optimal parameters of the following optimization problem:

$$\text{minimize}_{N_e, N_d} \quad \alpha N_e^{-p_e} N_d^{-p_d} \atop \text{s.t.} \quad N_e + N_d \leq B \tag{2}$$

To convert the problem to a convex problem, we instead consider the log of the objective and adopt the following change of variables:

$$u \equiv \log(N_e), \qquad v \equiv \log(N_d). \tag{3}$$

The transformed optimization problem is of the form:

$$\text{minimize}_{u,v} \quad -p_e u - p_d v \atop \text{s.t.} \quad \exp(v) + \exp(u) \leq B \tag{4}$$

Note that (4) is now convex and therefore, we can use KKT conditions to solve for the optimum. The Lagrangian has the form:

$$\mathcal{L}(u, v, \lambda) = -p_e u - p_d v + \lambda \bigg( B - \exp(v) - \exp(u) \bigg). \tag{5}$$

Solving for the first-order conditions yield:

$$-p_e = \lambda \exp(u^*) \tag{6}$$
$$-p_d = \lambda \exp(v^*). \tag{7}$$

Since the constraint is binding, $\lambda \neq 0$. Therefore, we can divide both sides of the equations above which yields:

$$\frac{p_e}{p_d} = \frac{\exp(u^*)}{\exp(v^*)} = \frac{N_e^*}{N_d^*}. \tag{8}$$

Substituting (8) in the constraint yields:

$$N_e^* = \frac{p_e}{p_e + p_d} B, \qquad N_d^* = \frac{p_d}{p_e + p_d} B. \tag{9}$$

Finally, we substitute (9) in the scaling law which yields:

$$\hat{L}_{opt}(B) = \alpha \left( \frac{p_e + p_d}{p_e B} \right)^{p_e} \left( \frac{p_e + p_d}{p_d B} \right)^{p_d} + L_\infty \tag{10}$$

$$= \alpha \left( \frac{p_e + p_d}{p_e} \right)^{p_e} \left( \frac{p_e + p_d}{p_d} \right)^{p_d} B^{-(p_e + p_d)} + L_\infty \tag{11}$$

$$\square$$

# H    OTHER SCALING APPROACHES

In the main text, we focus our empirical analysis on depth scaling – we increase the depth of the encoder or the decoder while keeping the model dimension (1024), the width of the feed-forward layer (8196), and the number of attention heads (16) fixed. In this section, we examine the role of the scaling approach in our results. We consider the following scaling approaches:

*Width Scaling*: The number of encoder/decoder layers is fixed at six. The model dimension increases from 768 all the way to 4096.

*Depth Scaling - Wide*: The number of encoder/decoder layers changes from 3 all the way to 32. Model dimension is fixed at 2048.

*Depth Scaling - Narrow*: The number of encoder/decoder layers changes from 3 all the way to 32. Model dimension is fixed at 768.

*Uniform Scaling*: Model depth, model dimension, feed-forward layer size, and number of attention heads are increase in tandem together.

Similar to the models in the main text, we use a dropout of 0.1 for residuals, feed-forward activations and attention. Models are trained with label smoothing of magnitude 0.1. To improve the training stability, all models use logit clipping of 10. We use Adafactor (Shazeer & Stern, 2018) optimizer for training our models. We use 40k linear warm-up steps and an inverse square root learning rate schedule. For Adafactor we used momentum with 0.9 and factored second moment to save memory.

Table 6 provides a detailed description of the models used in this analysis. In total, we consider 22 models with sizes ranging from 170M parameters all the way to nearly 4B parameters. For the sake of computational tractability, we only focus on symmetrically scaled models.

We examine how much the scaling behavior of these models deviates from the scaling trends computed in the main text. Figure 23 presents the percentage deviation of each model from the scaling law, computed as

$$\text{Percentage Deviation} = \frac{\text{Empirical Loss - Predicted Loss}}{\text{Empirical Loss}} \times 100. \tag{12}$$

The results suggest that the different scaling approaches seem to yield very similar scaling behaviors. In fact, for most of our models, the deviation from the scaling laws of the main text are limited to $1 - 2$ percents. This suggests that the phenomena identified in the main text are robust to the scaling approach.

The results of Figure 23 suggest that as we move to smaller models (less than 100M non-embedding parameters) the deviations from the scaling law seem to grow. We believe two factors contribute to this phenomena: (i) Model shape plays a key role in the final performance of small scale models. The influence of model shape diminishes as model size increases. (ii) Our scaling laws are parametrized to focus on the large model regime. Given that our scaling law is singular when either $N_e$ or $N_d$ are equal to zero, we do not anticipate getting high accuracy predictions from them in the small model regime.

Table 6: Description of the models used for alternative scaling.

| Model Class | #Layers | Model Dim | FF Width | # Heads | Total Parameters |
|---|---|---|---|---|---|
| Width Scaling | 6L6L | 768 | 8192 | 16 | 267M |
| Width Scaling | 6L6L | 1024 | 8192 | 16 | 375M |
| Width Scaling | 6L6L | 1536 | 8192 | 16 | 620M |
| Width Scaling | 6L6L | 2048 | 8192 | 16 | 902M |
| Width Scaling | 6L6L | 2560 | 8192 | 16 | 1221M |
| Width Scaling | 6L6L | 4096 | 8192 | 16 | 2407M |
| Depth Scaling-Wide | 3L3L | 2048 | 8192 | 16 | 549M |
| Depth Scaling-Wide | 6L6L | 2048 | 8192 | 16 | 902M |
| Depth Scaling-Wide | 8L8L | 2048 | 8192 | 16 | 1137M |
| Depth Scaling-Wide | 12L12L | 2048 | 8192 | 16 | 1607M |
| Depth Scaling-Wide | 16L16L | 2048 | 8192 | 16 | 2077M |
| Depth Scaling-Wide | 24L24L | 2048 | 8192 | 16 | 3016M |
| Depth Scaling-Wide | 32L32L | 2048 | 8192 | 16 | 3957M |
| Depth Scaling-Narrow | 3L3L | 2048 | 8192 | 16 | 171M |
| Depth Scaling-Narrow | 6L6L | 2048 | 8192 | 16 | 267M |
| Depth Scaling-Narrow | 8L8L | 2048 | 8192 | 16 | 332M |
| Depth Scaling-Narrow | 12L12L | 2048 | 8192 | 16 | 461M |
| Depth Scaling-Narrow | 16L16L | 2048 | 8192 | 16 | 590M |
| Depth Scaling-Narrow | 24L24L | 2048 | 8192 | 16 | 848M |
| Depth Scaling-Narrow | 32L32L | 2048 | 8192 | 16 | 1107M |
| Uniform Scaling | 6L6L | 768 | 3072 | 12 | 173M |
| Uniform Scaling | 12L12L | 1024 | 4096 | 16 | 451M |
| Uniform Scaling | 12L12L | 1536 | 6144 | 16 | 941M |
| Uniform Scaling | 12L12L | 2048 | 8192 | 24 | 1604M |
| Uniform Scaling | 16L16L | 2560 | 10240 | 32 | 3183M |

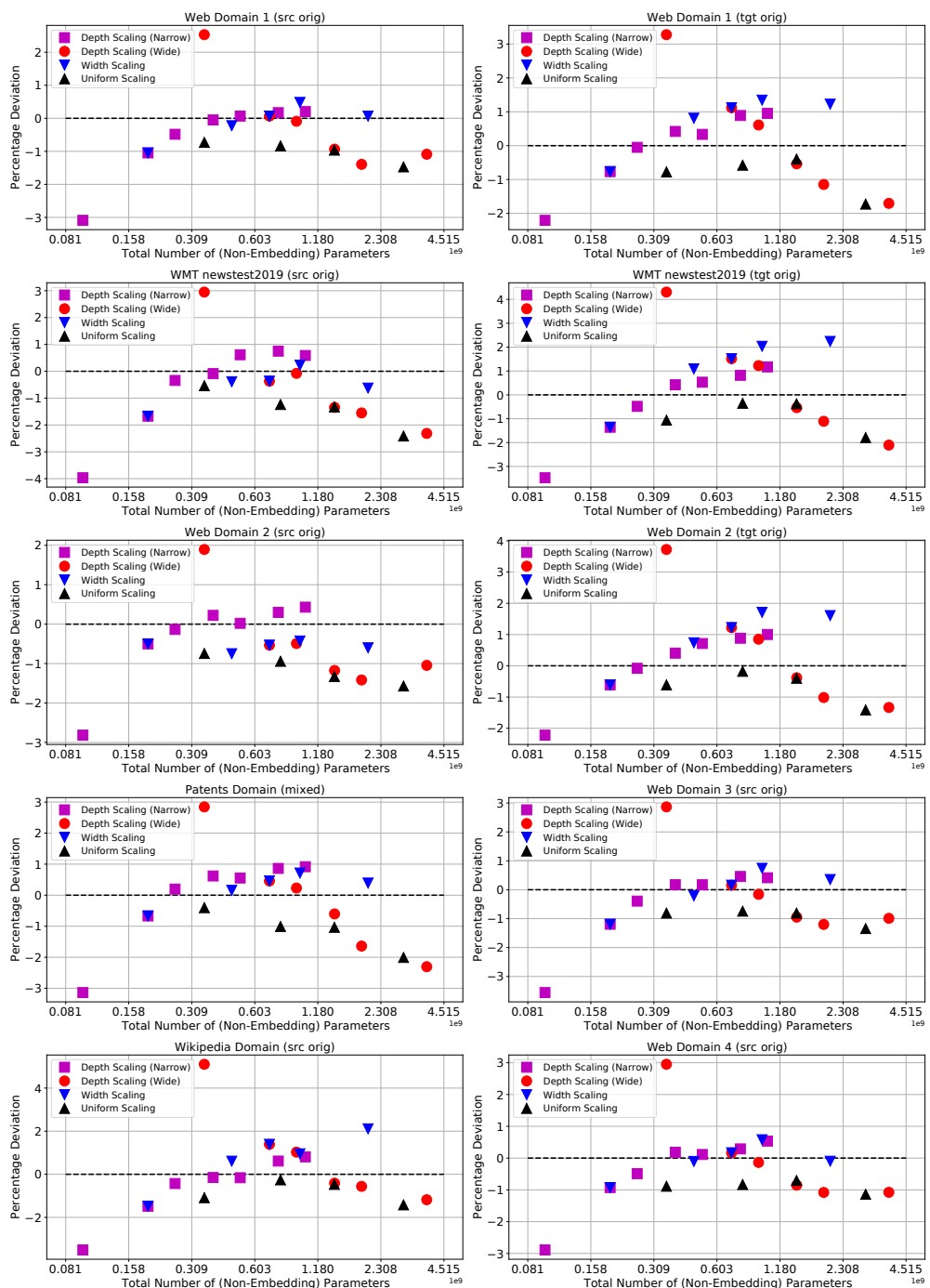

Figure 23: Percentage deviation from the scaling laws fitted to the English-to-German models presented in the main text. Different scaling approaches seem to yield similar results; for the majority of our models, different scaling approaches deviate only $1 - 2\%$ from the scaling trends presented in the main text. Deviation from the trend increases for smaller models suggesting that model shape plays an important role in the performance of small scale models. Models that experienced severe training instabilities / NaNs during training are removed from the plots.

# I   CURVE FITTING DETAILS

We use `scipy.optimize.least_squares` function for curve fitting throughout this paper [10]. To have some robustness to outliers, we use the `loss='soft_l1'` option which is a popular option for robust regression. The code snippet below shows the exact arguments we use for fitting the scaling laws:

```python
def func(p, x, y):
  """Fitting a bivariate scaling law.

  p: A 1-D array of dim 4, corresponding to alpha, p_e, p_d, c.
  x: A matrix of dimension n \times 2. First column encoder params,
    second col decoder params.
  y: A 1-D array of log-pplx of dim n."""
  x_e = NE_bar / x[:, 0]
  x_d = ND_bar / x[:, 1]
  return p[0] * np.power(x_e , p[1]) * np.power(x_d , p[2]) + p[3] - y

def fit_model(x, y, f_scale):
  X = x.to_numpy().copy()
  y = y.to_numpy().copy()
  if np.isnan(X).any() or np.isnan(y).any():
    raise ValueError('Data contains NaNs')
  if len(y.shape) > 1 or y.shape[0] != X.shape[0]:
    raise ValueError('Error in shapes')

  p0 = np.zeros((4,))
  p0[0] = 0.2 # alpha
  p0[1] = 0.4 # p_e
  p0[2] = 0.6 # p_d
  p0[3] = 1.0 # c
  fit = least_squares(func, p0, loss='soft_l1', f_scale=f_scale,
                      args=(X, y), max_nfev=10000, bounds=(0, 10))
  return fit
```

The `'soft_l1'` loss chosen above applies $\ell_2$ penalty on small residuals and a $\ell_1$-like penalty on outlier residuals. The argument `f_scale` determines the boundary where the transition between the two different behaviors occur. For the results presented in this paper, we choose `f_scale` from the grid given by `np.geomspace(0.001, 0.025, num=25)`. Choosing `f_scale=0.025` effectively yields a least-squares regression while smaller values add more robustness to outliers.

---

[10]`https://docs.scipy.org/doc/scipy/reference/generated/scipy.optimize.least_squares.html`

## J    ANALYSIS OF THE GENERATION QUALITY

**Decoding:**    As described in Section 4, we use beam-search for decoding (Wu et al., 2016). To keep the experiments tractable, we did not attempt to tune the hyper-parameters of beam-search for each model. Instead, we use the same hyper-parameters (optimized for the baseline model) for all our decoding jobs. In particular, we fix the length normalization parameter to $1.0$ and number of beams to $4$.

**BLEU-Cross Entropy Loss Co-Evolution:**    Figure 24 presents the relationship between BLEU score and cross-entropy loss for various test datasets. The results closely mimic the phenomenon observed in Figure 8: On target original data, improvements to cross-entropy loss are accompanied with improvements in BLEU score. On source original data however, beyond a certain point, cross-entropy loss and BLEU score exhibit diverging behaviors.

We observe that in large well-trained models, the relationship between BLEU and cross-entropy loss on target-original data is well captured by a power law of the form

$$\text{BLEU} = c_B L^{-p_B}, \qquad c_B, \ p_B > 0. \tag{13}$$

The fit achieved by this power law is plotted in our figures. We observe that fitted power laws for encoder scaling models consistently attain larger exponents compared to decoder or symmetrically scaled models. This reflects the fact that encoder scaling models are more successful in improving the generation quality (as measured by BLEU).

Finally, we observe a number of deviations from the predictions of Eq. (13). In particular, models with shallow decoders (6L2L, 6L4L, 6L6L) seem to outperform the trend (Figure 25). Moreover, we observe that models in the beginning of the training process tend to deviate from the overall trend (Figure 26). We postpone an in-depth analysis of these phenomena to future work.

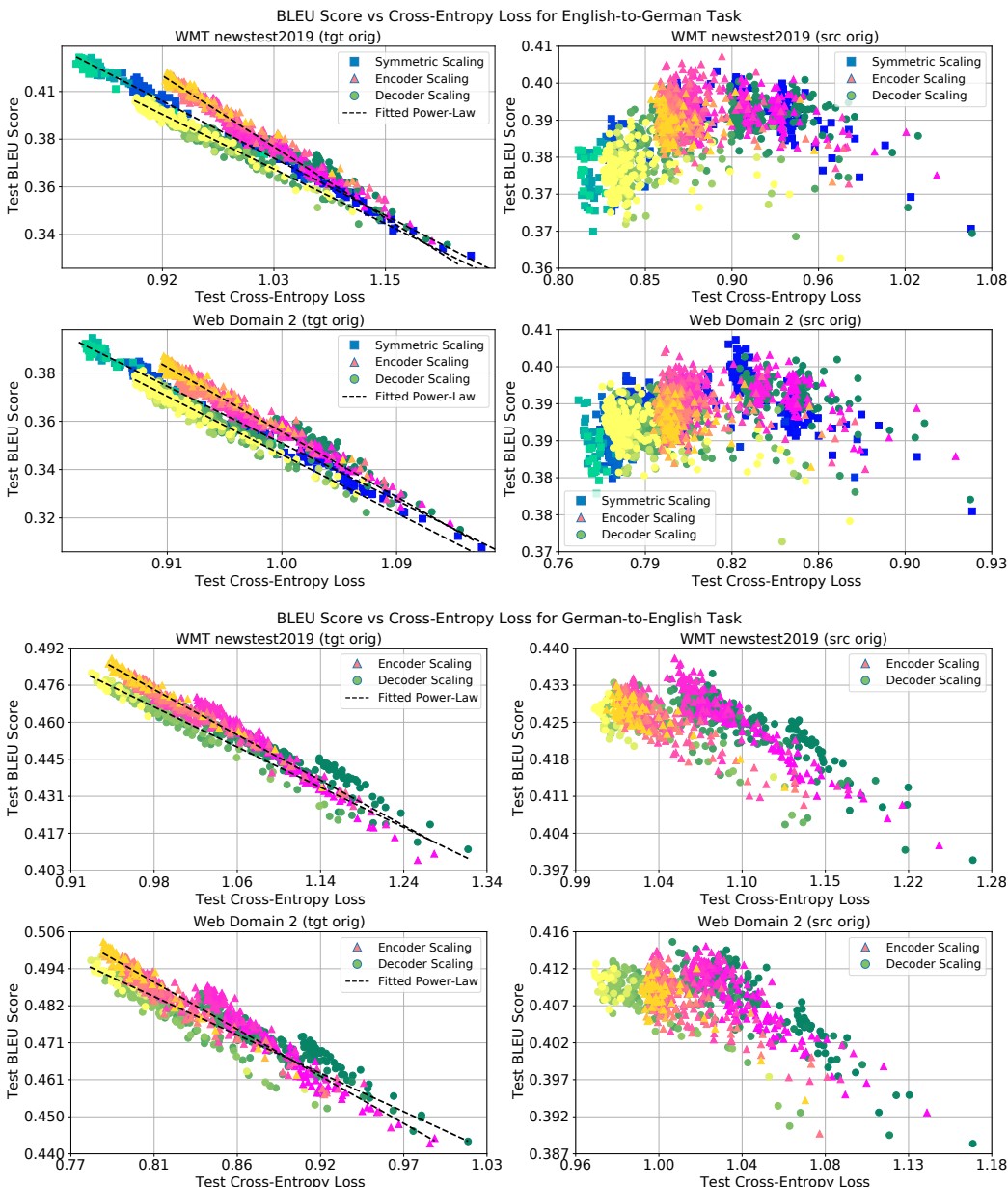

Figure 24: Log-log plot of the evolution of BLEU score as a function of cross-entropy loss for different models. For each scaling approach, warmer colors represent larger models. Each individual color represents different checkpoints of a single model during training. On target original data (left column), improvements to cross-entropy loss lead to consistent improvements in BLEU score. Dashed lines correspond to fit achieved by Eq. (13). The relationship breaks down for source original data (right column).

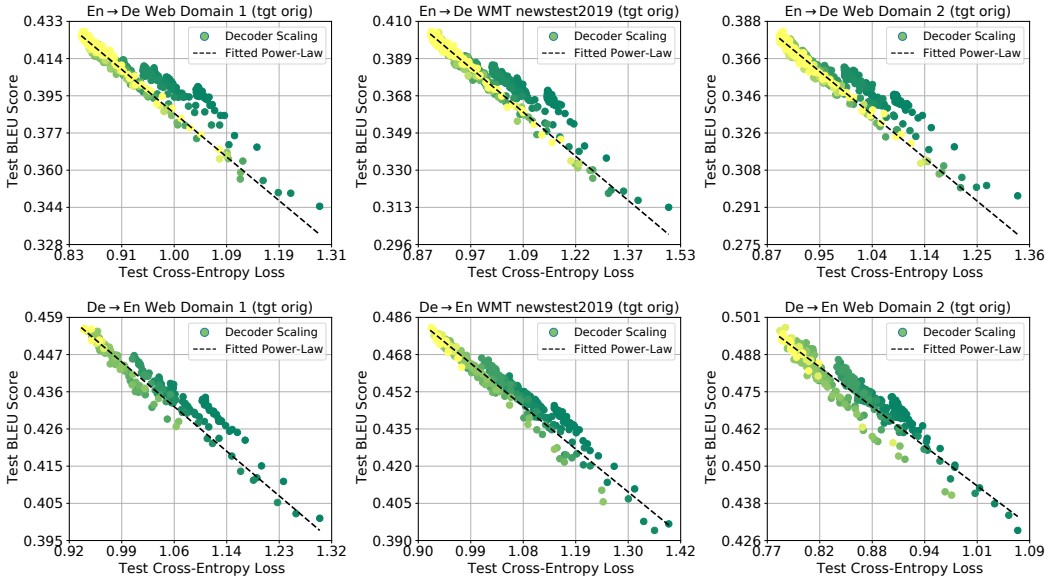

Figure 25: Models with shallow decoders tend to outperform predictions of Eq. (13). Points with dark green color represent different checkpoints of 6L2L, 6L4L, and 6L6L models.

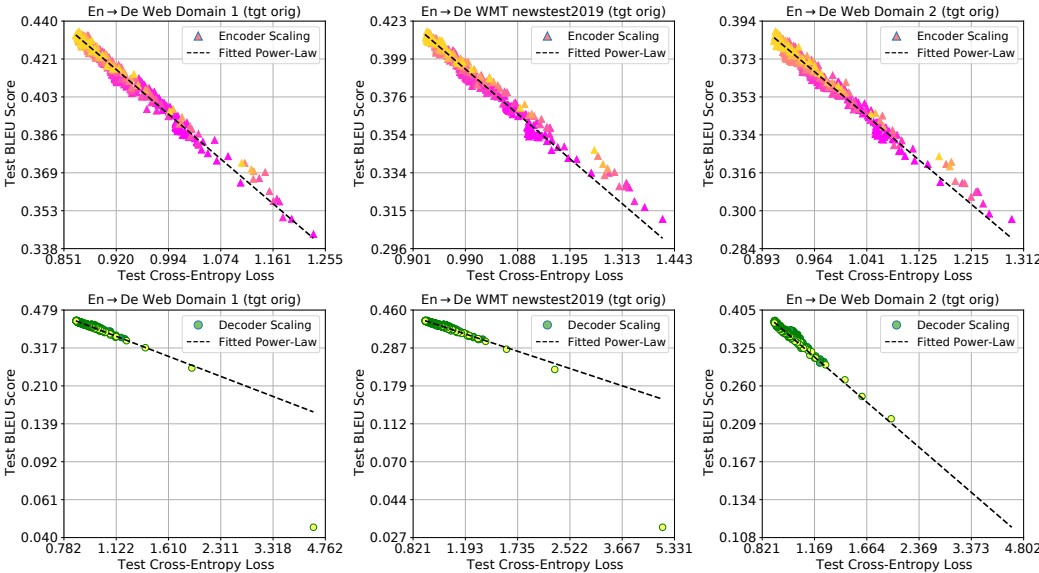

Figure 26: On some of the test sets, data points corresponding to early training checkpoints exhibit deviations from the overall trend.

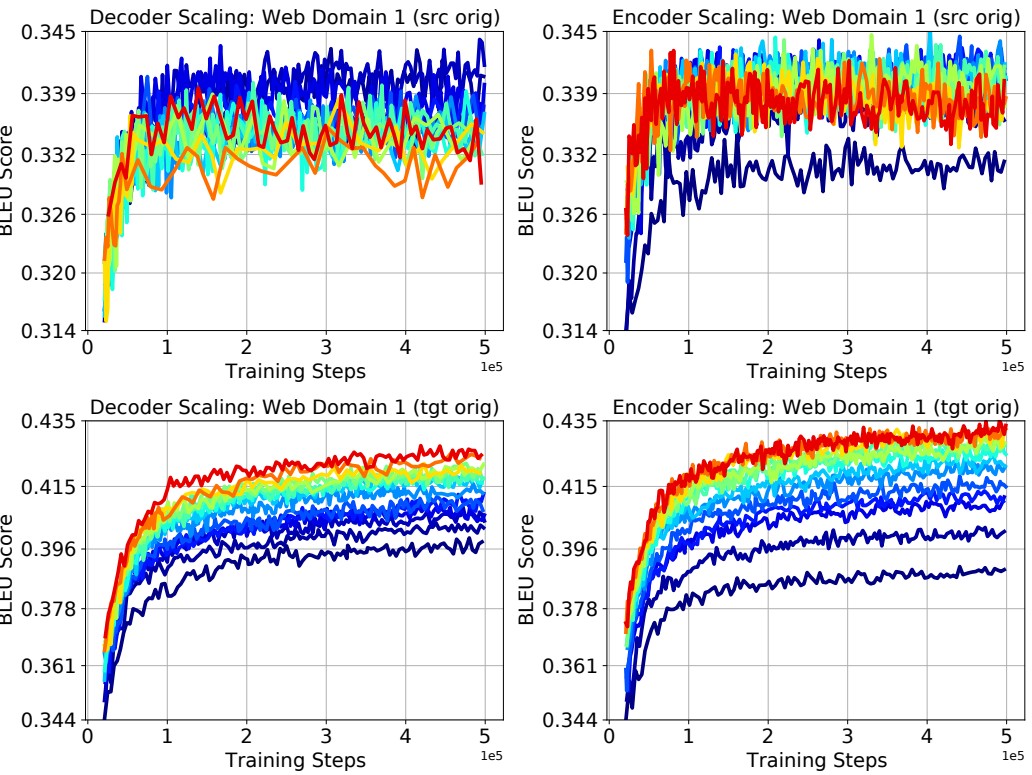

Figure 27: The evolution of BLEU score during the training for English-to-German Web Domain test sets. Warmer colors correspond to larger models. Top row: On source original test data, our largest models achieve lower BLEU scores compared to mid-sized models throughout the training. Bottom row: On target original test data, increasing the model size yields consistent improvements in BLEU score throughout the training.

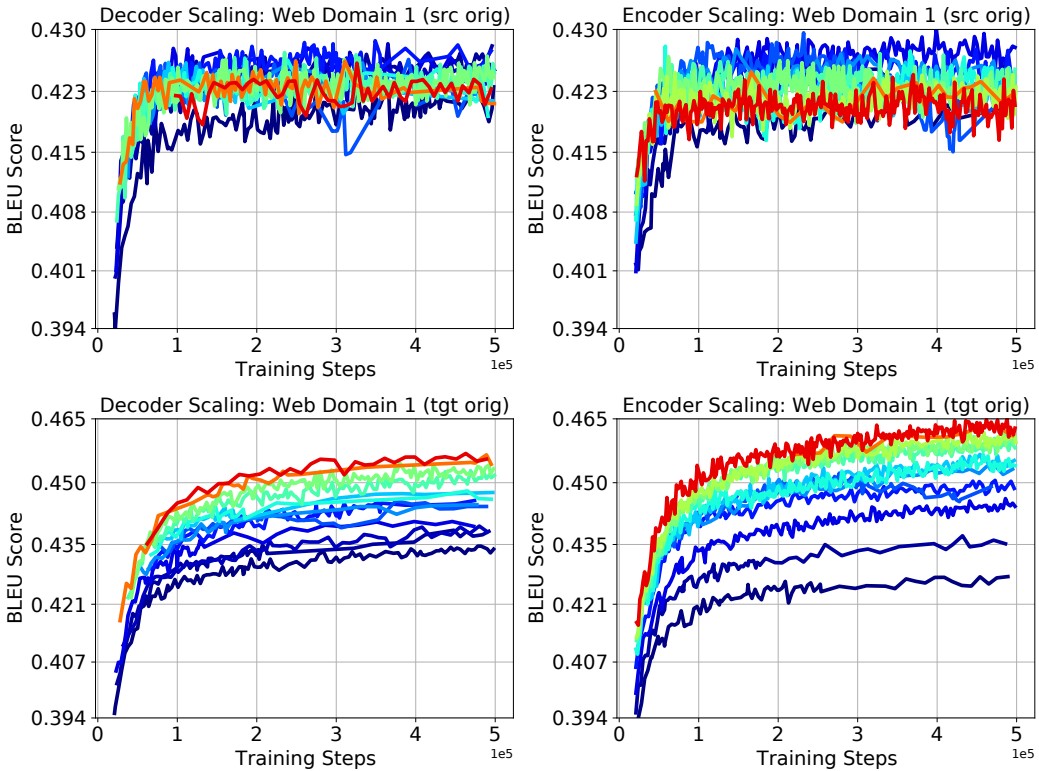

Figure 28: The evolution of BLEU score during the training for German-to-English Web Domain test sets. Warmer colors correspond to larger models. Top row: On source original test data, our largest models achieve lower BLEU scores compared to mid-sized models throughout the training. Bottom row: On target original test data, increasing the model size yields consistent improvements in BLEU score throughout the training.

# K  BLEURT SCORES

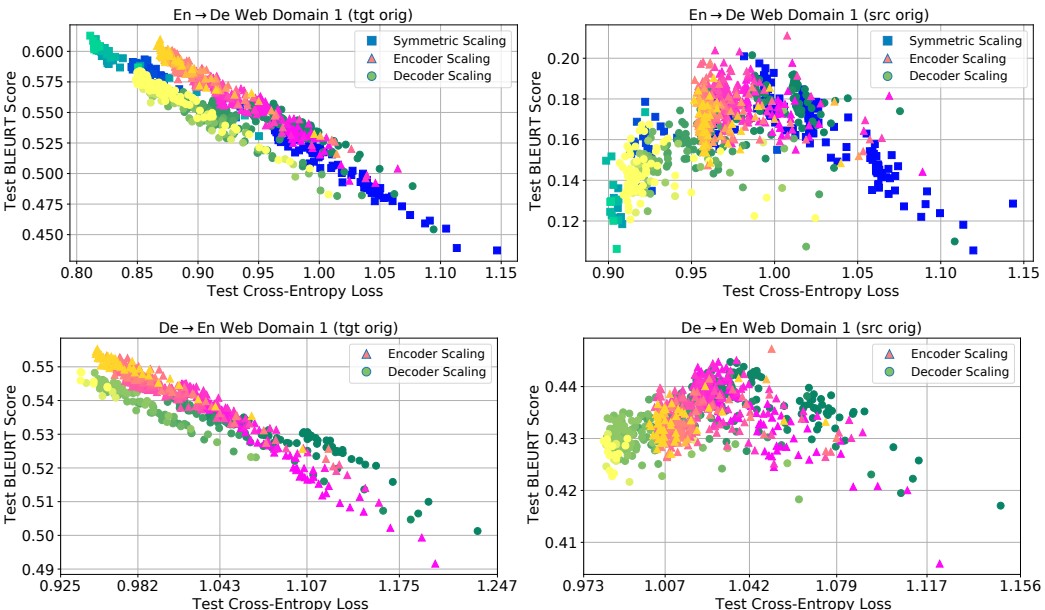

Figure 29: The evolution of BLEURT score as a function of cross-entropy loss for different models. For each scaling approach, warmer colors represent larger models. Each individual color represents different checkpoints of a single model during training. On target original data (left column), improvements to cross-entropy loss lead to consistent improvements in BLEURT score. This relationship breaks down for source original data (right column).

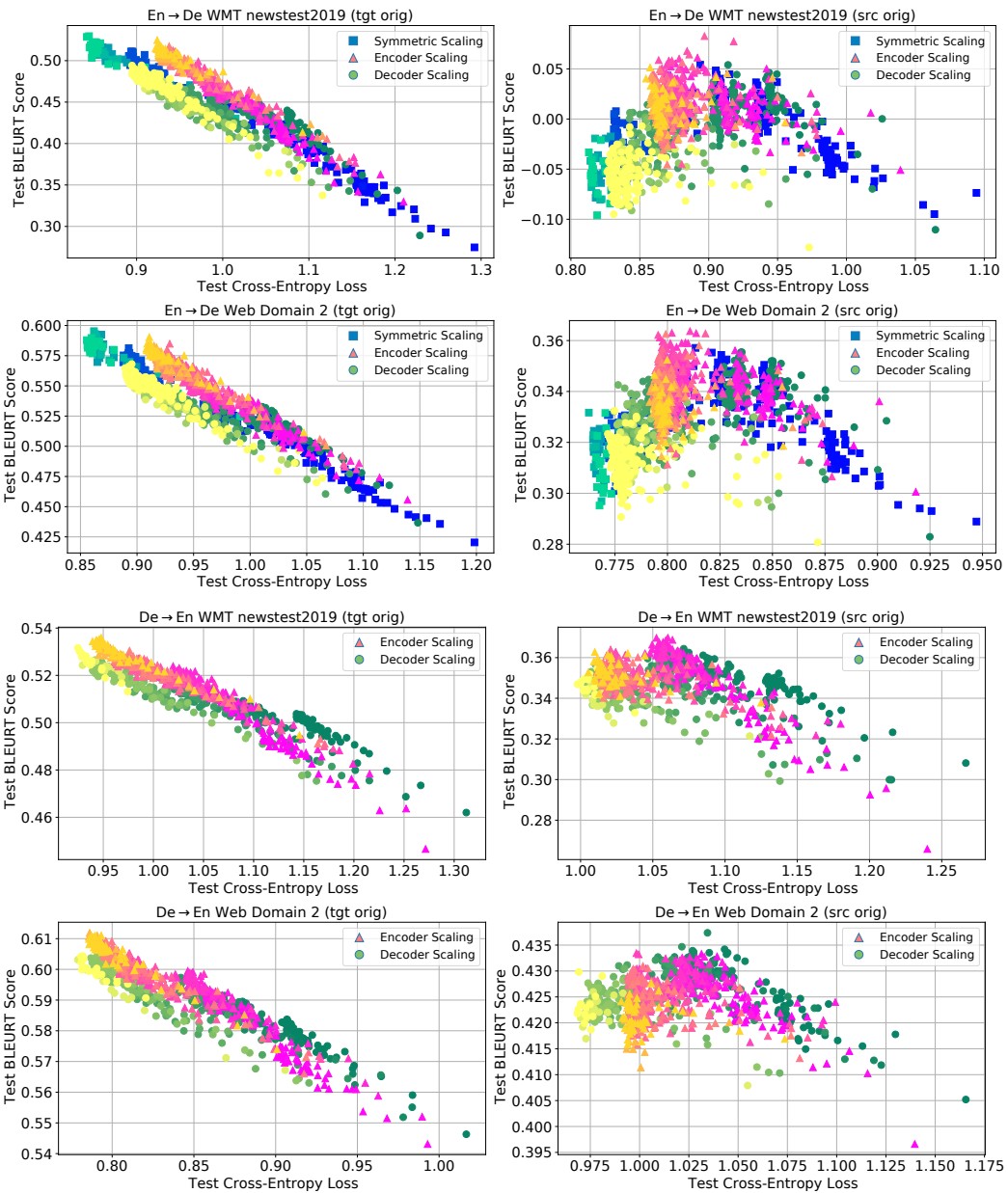

Figure 30: The evolution of BLEURT score as a function of cross-entropy loss for different models. For each scaling approach, warmer colors represent larger models. Each individual color represents different checkpoints of a single model during training. On target original data (left column), improvements to cross-entropy loss lead to consistent improvements in BLEURT score. This relationship breaks down for source original data (right column).

