# OpenReview forum: "Scaling Laws for Neural Machine Translation"
_ICLR.cc/2022/Conference — ICLR 2022 Spotlight_

### Official Review · Reviewer_TAXy · 2021-11-01

**Correctness:** 4
**Technical Novelty And Significance:** 2
**Empirical Novelty And Significance:** 4
**Recommendation:** 8
**Confidence:** 4

**Main Review:**

Strengths:
1. They conduct extensive experiments on four language pairs and five different test domains.
2. It is interesting that contrary to previous findings, this paper suggests that it is better to scale the decoder than the encoder, providing guidance for future research on deep NMT.
3. The analysis on the translationese provides insights on the correlations between cross-entropy loss and generation quality, and its effect on the scaling laws.

Weaknesses:
1. The scaling laws have several test set-specific parameters, which makes it questionable if the laws have practical values. For example, if we want to get the optimal allocation of encoder/decoder capacity using their method, we have to train models of different variants and fit their proposed formula, so why don’t we just do parameter search with the same training budget?
2. I appreciate that the authors make efforts to measure the outputs in both BLEU and BLEURT, but it would be better to include human evaluation results that can better reflect the generation quality of models.
3. The paper mainly focuses on the relationship between cross-entropy loss and the number of Transformer layers in large-scale settings, while there are a lot of other factors that can be considered. For example, it is unclear whether the scaling laws will hold if the architecture varies or if the hidden size changes.


**Summary Of The Paper:**

The paper provides scaling laws for machine translation models. They experiment on different language pairs and domains and focus on large-scale Transformer-based NMT models.

They find that the scaling behavior of cross-entropy loss can be described as a bivariate function of encoder and decoder size and make recommendations for optimal allocation of encoder/decoder capacity. They also find that “translationese” can affect the model scaling behavior. Finally, they study the correlations between cross-entropy loss and BLEU/BLEURT scores.

**Summary Of The Review:**

While the paper has some minor issues, it conducts extensive experiments in large-scale settings and the results can provide insights for future research, thus I’m leaning towards an acceptance of the paper

---

> ### Author Response · Authors · 2021-11-17
> **Response to Reviewer TAXy**
>
> We thank the reviewer for their insightful comments. We address the reviewer's concerns below:
>
> 1. Indeed, to fit the parameters of the scaling law, one needs to train a number of models. However, in practice, one can use smaller models to fit the scaling law and then make predictions for much larger models. As an example, in Figure 3, we show that scaling laws fitted using models with 100M-1.5B parameters can almost perfectly predict the performance of models with $\approx 3.5B$ parameters. To take this example to the extreme, for image classification and language modeling tasks, [Rosenfeld et al (2019)](https://arxiv.org/abs/1909.12673) show that scaling laws fitted using smaller models can be used to accurately predict the performance of models 16 to 32 times larger (see Section 7 of that paper and the discussion within). This makes the scaling law approach a computationally appealing approach for designing and prototyping massive models (with potentially O(100B) parameters). In contrast, due to heavy computational costs of training such massive models, naive parameter searches are prohibitively expensive and infeasible.
>
>     In addition, we emphasize that many of the insights we provide in the paper do not rely on having the test set-specific scaling coefficients. In particular, as Figure 10 (left) demonstrates, $\frac{N_d}{N} \approx 0.7$ parameter allocation scheme is approximately optimal **universally across all test sets**. Similarly, the insights we provide regarding the train & test data composition biases can be leveraged without any computational effort.
>
> 2. We fully agree with the reviewer on the importance of examining human evaluation results. Considering the many recently reported intricacies of MT evaluation ([Freitag et al 2021](https://arxiv.org/abs/2104.14478); [Kocmi et al 2021](https://arxiv.org/abs/2107.10821)), and the complex interactions that emerge between search errors and model fit with scaling, we decided to postpone such an exploration to a dedicated follow-up study where different aspects of the problem can be studied in-depth.
>
>     Moreover, as an added step to allow for further investigation in this direction, we release the texts generated by our models so that the research community can inspect and examine the output of our models beyond the quality metrics reported in the paper.
>
> 3. Indeed one might be concerned that changing the architecture (number of heads, hidden dimension size, …) might drastically affect the scaling law. To investigate this, in Appendix G, we examine a number of other scaling approaches (width scaling, depth scaling with different model widths, uniform scaling of width / depth / and attention heads, …). We observe that the final performance of these models shows just 1-2% deviation from the predictions of our scaling law. As such, it seems that while the model shape and hyper-parameters are important, as long as they are chosen in a reasonable range, their effect is limited. This observation closely mirrors [Kaplan et al (2020)](https://arxiv.org/abs/2001.08361) observation that when scaling language models, “Performance depends very mildly on model shape when the total number of non-embedding parameters N is held fixed” (See Figure 5 of that paper).
>
> References:
> - Rosenfeld, Jonathan S., et al. "A Constructive Prediction of the Generalization Error Across Scales." International Conference on Learning Representations. 2019.
> - Freitag, Markus, et al. "Experts, errors, and context: A large-scale study of human evaluation for machine translation." arXiv preprint arXiv:2104.14478 (2021).
> - Kocmi, Tom, et al. "To ship or not to ship: An extensive evaluation of automatic metrics for machine translation." arXiv preprint arXiv:2107.10821 (2021).
> - Kaplan, Jared, et al. "Scaling laws for neural language models." arXiv preprint arXiv:2001.08361 (2020).

---

> > ### Comment · Reviewer_TAXy · 2021-11-22
> > **Thanks for the clarifications**
> >
> > The response addresses most of my concerns. I've increased my score and look forward to the human evaluations.

---

### Official Review · Reviewer_afSG · 2021-11-01

**Correctness:** 4
**Technical Novelty And Significance:** 4
**Empirical Novelty And Significance:** 4
**Recommendation:** 10
**Confidence:** 5

**Main Review:**


Strengths:
- the research questions answered in this paper are interesting and help to better understand the mechanics of NMT. As an expert in MT, I really found this paper interesting.
- findings are very insightful
- this paper will be impactful: it will guide NMT  (or any other kind of encoder-decoder model such a BART) practictionners to build more efficient and effective models
- another proof that target-original test sets are not suitable for NMT evaluation (Figure 8 should go on Twitter when you publish your paper!)
- evaluation is not only performed with BLEU (BLEURT is also used)


Weaknesses:
- almost the entire evaluation is not reproducible: undisclosed datasets, use of a non-standard BLEU implementation, pre-processing of datasets not detailed, etc.
- the paper misses a related work section (more details than in the introduction). The reader may not know related work on language models for instance

**Summary Of The Paper:**

This paper presents a study on the scaling power of NMT Transformer (encoder-decoder).

It shows:
- how the scaling of encoder-decoder compares to language model
- that NMT behaves very differently on translationese texts
- propose to model the scaling of NMT encoder-decoder


**Summary Of The Review:**

All the "strengths" pointed above justifies my recommendation for the paper.

---

> ### Author Response · Authors · 2021-11-23
> **Response to Reviewer afSG**
>
> We thank the reviewer for their supportive feedback.
>
> 1. We fully agree with the reviewer on the importance of reproducibility. For this study, we decided to use a large high-quality in-house dataset to ensure that training data size will not bottleneck the learning dynamics (so that the effects of model scaling can be clearly observed). We aim to switch to public datasets for future studies as more and more large-scale public datasets (such as ParaCrawl) become available.
>
> 2. We have expanded the discussion of the relevant literature in the revised version (Appendix A) and have commented on how our investigation relates to recent results in the scaling laws literature.

---

### Official Review · Reviewer_q9c2 · 2021-11-01

**Correctness:** 4
**Technical Novelty And Significance:** 3
**Empirical Novelty And Significance:** 3
**Recommendation:** 8
**Confidence:** 4

**Main Review:**

Pros:
1. The proposed function of model capacity for encoder-decoder NMT model performance is useful for parameter allocation of NMT models training.
2. This paper provides experiments to explore how the composition of training/test data affects the test/training loss for scaling NMT models. It's an important question to connect the other two questions.
3. It is important to investigate how generation quality evolves with the improvements in cross-entropy resulting from model scaling. The problem is of great value for both NMT practice and research.


Cons:
1. The authors have proposed three research questions, while the relationship among them is not clear.
2. The scaling property of the composition bias of the train/test sets has been less studied in previous work, and is most interesting to the reviewer. This part can be more convincing if the authors can provide more experiments and analyses.

**Summary Of The Paper:**

The paper provides interesting scaling properties for autoregressive neural machine translation models, which are validated on comprehensive experiments. The authors investigate the scaling properties from three perspectives:
1. Quantify the evolution of model quality as a function of model capacity for encoder-decoder NMT models.
2. How the composition of training/test data affects the scaling behavior of NMT models?
3. How generation quality evolves with the improvements in cross-entropy resulting from model scaling?

**Summary Of The Review:**

The research questions explored in the study are important and interesting, e.g., the scaling properties of composition bias and the influence of source/target naturalness on the scaling behaviors. However, the relationship among these research questions and findings is not clear.

---

> ### Author Response · Authors · 2021-11-17
> **Response to Reviewer q9c2**
>
> We thank the reviewer for their comments. We address the reviewer's concerns below:
>
> 1. In this work, we aimed to understand how increasing the model size affects the generalization performance in NMT. In our study, we observed three elements confounding the scale-generalization relationship
>     1. **Model Architecture:** Are parameters being added to the encoder or the decoder?
>     2. **Data Composition:** Are the source / target sides natural human generated text?
>     3. **Inference Step:** Is our inference routine capable of fully translating improvements in test cross-entropy loss to improvements in translation quality?
>
>     The three research questions raised on page 2 each isolate and study one of these confounding factors. The answer to these questions paints a comprehensive picture of the scaling behavior in NMT.  We hope our clarification answered the reviewer's concern. If there is any specific aspect of the text that is causing confusion or if there are any suggestions for revision, we would be happy to address them.
> 2. Indeed, we agree with the reviewer that the study of composition bias merits further investigation; our experiments raise interesting questions regarding machine generated content and data naturalness that are of high interest to the community. Given the 9 page limit of the paper, there was simply no space left to examine and answer all these questions in sufficient detail here. We hope to point out and share these phenomena with the research community in this paper and explore them in more depth in follow-up studies.

---

> > ### Comment · Reviewer_q9c2 · 2021-11-23
> > **Thanks for the response.**
> >
> > I have read other reviewer' comments and all the responses. I've raised my score for the detailed responses and the informative revisions. Still, I hope to see more experiments on the composition bias in the revised version.

---

### Official Review · Reviewer_Dbxb · 2021-11-02

**Correctness:** 4
**Technical Novelty And Significance:** 3
**Empirical Novelty And Significance:** 3
**Recommendation:** 8
**Confidence:** 4

**Main Review:**

Pros:
- I really enjoyed reading this paper. The research questions are clearly stated, and the results are fascinating to go through. Well-done!
- The number of experiments is impressive. This adds weight to the papers conclusions, which are certainly thought-provoking. For example, the conclusion that scaling law should be bivariate, treating the number of encoder and decoder layers separately, makes me think we should not train models of symmetric number of layers anymore.

Cons:
- The footnote on comparison to Gordon et. al. in EMNLP2021 is too terse and should be expanded. That paper studied scaling laws for NMT as well, but in a different parameter regime. Do your conclusions affirm, reject, or extend theirs?

**Summary Of The Paper:**

This paper studies scaling laws for NMT. It confirms and extends some of the existing work on scaling laws, and as such it is a valuable contribution to the field. In particular, new questions addressed include: does the encoder-decoder architecture of NMT share the same scaling law as decoder-only LMs? How does training data (backtranslated or not) affect scaling?

**Summary Of The Review:**

This is a strong empirical paper that extends our understanding of scaling laws in deep learning. The research questions are clear and the results are convincing. I think it will be cited often in the future.

---

> ### Author Response · Authors · 2021-11-23
> **Response to Reviewer Dbxb**
>
> We thank the reviewer for their supportive feedback. Indeed due to the time and space constraints, we were not able to include a comprehensive discussion of the relevant literature (Gordon et al in particular) to the initial submission. We have expanded this discussion in our revision (Appendix A).
>
> Compared to Gordon et al, we study a more general setup. In particular, we consider a setting where the encoder / decoder can be scaled asymmetrically (which is a common practice in MT). In addition, we also examine the role of data composition in scaling which has not been studied before. When considering only the symmetric scaling setting (which was the setup in Gordon et al), our results tend to agree with Gordon et al to a large degree. In this scenario, our scaling law simplifies to $L(N) =\alpha N^{-p} + L_{\infty}$ where N is the total number of parameters. This agrees with the conclusions of Gordon et al. One difference between our results is the presence of the $L_{\infty}$ term in our scaling law. This term, which corresponds to the irreducible loss associated with the problem, does not appear in Gordon et al’s analysis. We suspect that this might be due to the small models used in that analysis which cannot statistically observe such asymptotic terms.
>
> In addition, Gordon et al observe consistent improvements in BLEU score as model size increases. In comparison, we unveil a more intricate relationship in our study: We show that when the test data is target-original, improvements in the test cross-entropy loss are correlated with improvements in generation quality (as measured by BLEU/BLEURT scores). However, when the test set is source-original, the correlation between cross-entropy loss and generation quality breaks down. In this setting, we observe that generation quality improves up to a certain threshold and then deteriorates.

---

> > ### Comment · Reviewer_Dbxb · 2021-11-23
> > **thanks for the response**
> >
> > Thanks! This is a great response and putting it in appendix seems appropriate.

---

> > > ### Comment · Reviewer_Dbxb · 2021-11-29
> > > **Citation**
> > >
> > > By the way, note the Gordon citation should be updated to EMNLP2021 since it is now published. See bibtex here: https://aclanthology.org/2021.emnlp-main.478/
> > > Also, since the title of your paper is very similar ("Scaling Laws for NMT" vs "Data and Parameter Scaling Laws for NMT"), it may be worth considering adjusting your title to make a stronger distinction. It is up to you, though.

---

### Decision · Program_Chairs · 2022-01-20

**Decision:**

Accept (Spotlight)

**Comment:**

This is a strong empirical paper that studies scaling laws for NMT in terms of several new aspects, such as the model quality as a function of the encoder and decoder sizes, and how the composition of data affects scaling, etc. The extensive empricial results offer new insights to the questions and provide valuable guidance for future research on deep NMT. The datasets used in the study are non-public, which may make it hard to reproduce the evaluation.